DOI: 10.1038/s41467-018-07122-z　　**OPEN**

# Phenotype loss is associated with widespread divergence of the gene regulatory landscape in evolution

Juliana G. Roscito[1,2,3,4], Katrin Sameith[1,2,3], Genis Parra[1,2,3], Bjoern E. Langer[1,2,3], Andreas Petzold[5], Claudia Moebius[1], Marc Bickle [1], Miguel Trefaut Rodrigues[4] & Michael Hiller[1,2,3]

Detecting the genomic changes underlying phenotypic changes between species is a main goal of evolutionary biology and genomics. Evolutionary theory predicts that changes in *cis*-regulatory elements are important for morphological changes. We combined genome sequencing, functional genomics and genome-wide comparative analyses to investigate regulatory elements in lineages that lost morphological traits. We first show that limb loss in snakes is associated with widespread divergence of limb regulatory elements. We next show that eye degeneration in subterranean mammals is associated with widespread divergence of eye regulatory elements. In both cases, sequence divergence results in an extensive loss of transcription factor binding sites. Importantly, diverged regulatory elements are associated with genes required for normal limb patterning or normal eye development and function, suggesting that regulatory divergence contributed to the loss of these phenotypes. Together, our results show that genome-wide decay of the phenotype-specific *cis*-regulatory landscape is a hallmark of lost morphological traits.

---

[1] Max Planck Institute of Molecular Cell Biology and Genetics, Dresden 01307, Germany. [2] Max Planck Institute for the Physics of Complex Systems, Dresden 01187, Germany. [3] Center for Systems Biology Dresden, Dresden 01307, Germany. [4] Instituto de Biociências, Universidade de São Paulo, São Paulo 05508-090, Brazil. [5] Center for Regenerative Therapies TU Dresden, Dresden 01307, Germany. These authors contributed equally: Juliana G. Roscito, Katrin Sameith, Genis Parra. Correspondence and requests for materials should be addressed to M.H. (email: hiller@mpi-cbg.de)

Phenotypic diversity is most easily observable as differences in morphology. Understanding how morphological differences evolved is a central question in many areas of biology. Morphology is established during development and requires the patterning of the developing embryo to assign specific fates to cells. Patterning processes are controlled by genes that are often involved in the development of many different structures. Consequently, expression of these highly pleiotropic developmental genes must be tightly regulated in a spatio-temporal manner. Key to transcriptional regulation of pleiotropic developmental genes are *cis*-regulatory elements that can be located far upstream or downstream of the promoter and often control gene expression in specific tissues at specific time points. Thus, understanding how developmental genes and their *cis*-regulatory elements evolve is crucial to understanding how morphology evolves.

Differences in pleiotropy between developmental genes and *cis*-regulatory elements impact which mutations are permissible in evolution. Whereas mutations in the coding regions of a pleiotropic gene may affect gene function in many tissues, which is often deleterious, mutations in modular *cis*-regulatory elements likely affect gene expression only at a specific time and tissue. Accordingly, it was proposed that morphology largely evolves by changes in the spatio-temporal expression of developmental genes, which in turn evolves by changes in the underlying *cis*-regulatory elements[1,2]. This hypothesis is supported by a growing body of evidence [3–8].

The loss of a complex phenotype is one extreme case of morphological evolution. Upon phenotype loss, we expect a different evolutionary trajectory for the genetic information underlying this phenotype. On the one hand, the integrity of developmental genes should be maintained over time due to selection on those gene functions that are not related to the lost phenotype. On the other hand, modular *cis*-regulatory elements associated specifically with this phenotype may directly contribute to its loss and are expected to evolve neutrally afterwards. This should result in sequence divergence and thus decay of regulatory activity over time. These different trajectories are illustrated by the loss of pelvic spines in freshwater stickleback fish, caused by the loss of a pelvic-specific enhancer for the pleiotropic *Pitx1* gene, while *Pitx1* remained intact[4]. Similarly, mutations in the limb-specific ZRS enhancer for sonic hedgehog (*Shh*) in the snake lineage led to altered *Shh* expression in the limbs, while the pleiotropic *Shh* gene remained intact[7,8]. However, recent studies[8–10] found that numerous other limb enhancers are nevertheless still conserved in snakes, despite limb reduction in this lineage dating back to more than 100 Mya[11], possibly due to pleiotropy of regulatory elements that drive expression in other non-limb tissues. Thus, it remains an open question whether phenotype loss is generally associated with widespread divergence of the *cis*-regulatory landscape.

Here, we combine genome sequencing with functional and comparative genomics to systematically investigate the fate of *cis*-regulatory elements in lineages that lost complex phenotypes: loss of limbs in snakes and degeneration of eyes in subterranean mammals. Our analyses provide genome-wide evidence that divergence of the phenotype-specific *cis*-regulatory landscape is a hallmark of lost morphological traits. Furthermore, we present the first comprehensive picture of the non-coding genomic changes that likely contributed to limb loss and eye degeneration. More generally, our study provides a widely applicable comparative and functional genomics strategy to detect *cis*-regulatory element candidates that may underlie morphological differences between species.

## Results

**Sequencing and annotation of the tegu lizard genome.** We first investigated the fate of the limb-related *cis*-regulatory landscape in snakes. To detect sequence divergence that is specific to snakes, it is necessary to compare the genomes of snakes to the genomes of several fully limbed reptiles and other vertebrates. Given the sparsity of genomes of reptiles with well-developed limbs, we sequenced and assembled the genome of the fully limbed tegu lizard *Salvator merianae*, representing the first sequenced species of the teiid lineage. We used a combination of Illumina MiSeq and HiSeq sequencing (Supplementary Table 1), followed by iterative read error correction with SGA-ICE[12], and genome assembly with ALLPATHS-LG[13]. The resulting 2 Gb genome assembly has a contig N50 value of 176 Kb and a scaffold N50 value of 28 Mb, with the longest scaffold spanning 99 Mb. Compared with other sequenced reptiles, the tegu genome shows the largest contig N50 value and the second largest scaffold N50 value (Fig. 1a). The tegu assembly contains all of the 197 vertebrate non-exonic ultraconserved elements[14], and has a BUSCO[15] score of 96.8%, showing an assembly completeness higher than that of all other sequenced reptiles (Supplementary Table 2). Combining transcriptomics and homology-based gene prediction approaches, we annotated a total of 23,487 genes, 16,284 of which have a human ortholog.

**Genome alignment between snakes and limbed vertebrates.** Using the tegu genome as reference, we created a multiple genome alignment including two well-assembled snakes (boa and python), three other limbed reptiles (green anole lizard, dragon lizard, and gecko), three birds, alligator, three turtles, fourteen mammals, frog, and coelacanth (Fig. 1b; Supplementary Table 3). This alignment of 29 genomes was used as the basis for all further analyses.

**Divergence of conserved non-coding elements in snakes.** To study the evolution of *cis*-regulatory elements across species on a genome-wide scale, we focused on conserved non-coding elements (CNEs), because evolutionary sequence conservation often implies purifying selection and thus function, and because CNEs often overlap *cis*-regulatory elements[16]. Using our genome alignment, we identified a total of 164,422 non-coding elements (covering 1% of the tegu genome) that are conserved among many but not necessarily all Amniota species (workflow in Fig. 1c). For each CNE, we computed a per-species sequence divergence value by determining the percent of bases that are identical between the species' CNE sequence and the reconstructed sequence of the amniote ancestor[17,18]. By searching for CNEs that exhibit a substantially lower sequence identity in both boa and python snakes, we expect to identify *cis*-regulatory elements that control gene expression in the developing limbs of limbed species and that may no longer function as limb regulatory elements in limbless snakes. To identify such snake-diverged CNEs, we first computed a global Z-score, which measures how many standard deviations the sequence identity in snakes is below the average identity in all other limbed species. To assure that divergence is specific to snakes and to exclude relaxed selection in the entire reptile clade, we additionally computed a local Z-score by comparing the snakes and the three lizards alone. Requiring a Z-score cutoff of −3 for both comparisons, we identified 5439 CNEs that are highly and specifically diverged in snakes (Supplementary Table 4). Using fdrtool[19], we estimated that a Z-score cutoff of −3 corresponds to a false discovery rate (FDR) of 1.19%. We additionally obtained Z-score null distributions by simulating the evolution of CNEs under selection in

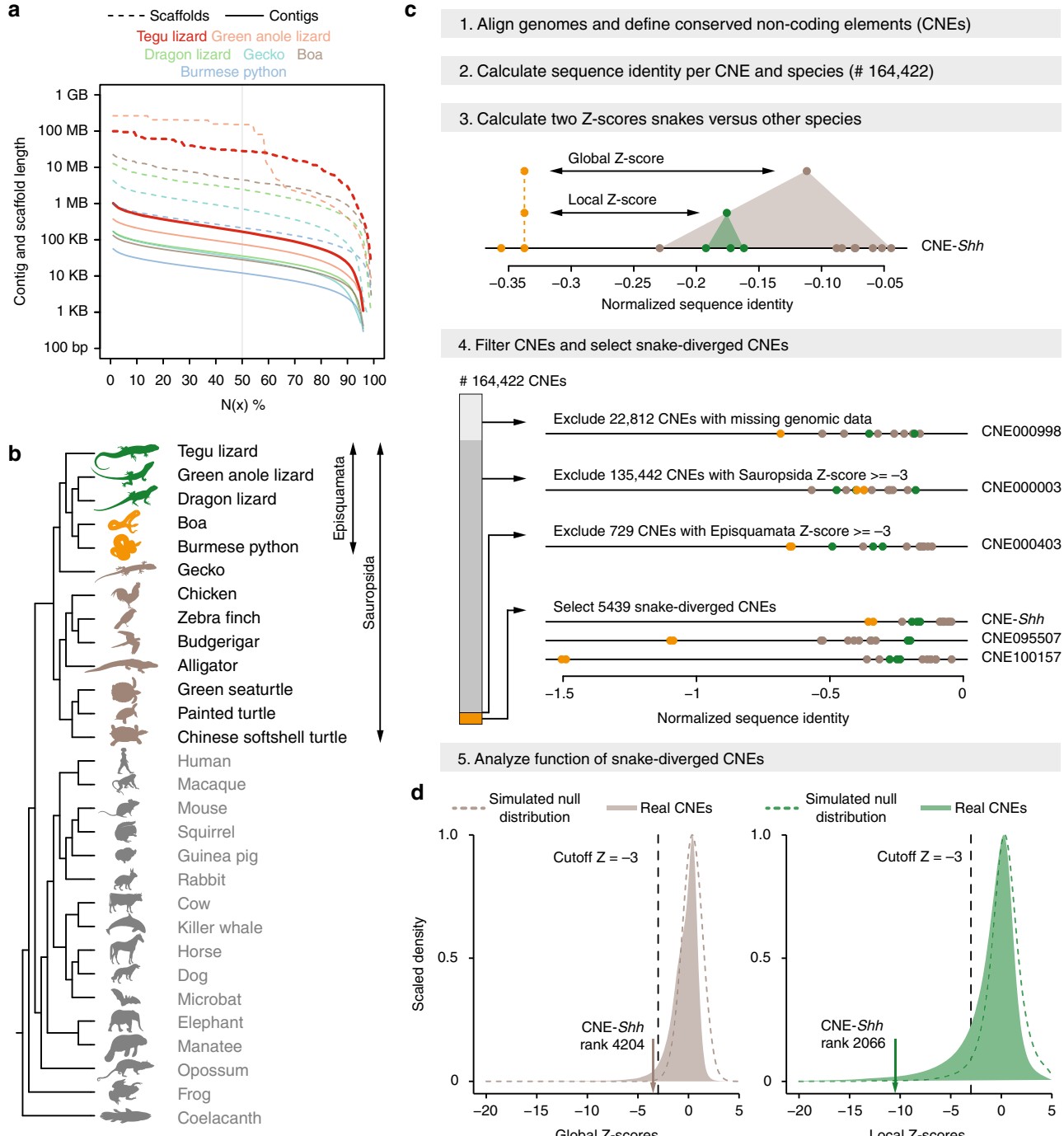

**Fig. 1** Comparative framework to detect sequence divergence of regulatory elements in snakes. **a** Comparison of the tegu genome to genomes of other sequenced reptiles. N(x)% graph showing the contig and scaffold size (*y*-axis), where x% of the genome consists of contigs and scaffolds of at least that size. **b** Phylogenetic tree of the limbless and limbed species included in our multiple genome alignment. Snakes are in orange, remaining Episquamata species in green, remaining Sauropsida species in brown, and outgroup species in gray. All animal illustrations were taken from phylopic.org. **c** Overview of the computational steps to identify CNEs that are specifically diverged in snakes. **d** Filled gray and green density curves depict global and local Z-score distributions of 141,610 CNEs, respectively (22,812 CNEs with missing genomic sequence are excluded). Dashed curves depict the Z-score null distributions obtained by simulating the evolution of CNEs under selection in all amniote species

all amniote species (Fig. 1d), which independently estimated the FDR to be 3.17%. Finally, to estimate an upper bound of the FDR, we detected 616 CNEs that are preferentially diverged in the sister lineage of snakes comprising the anole and dragon lizards. Making the unrealistic assumption that all 616 CNEs evolve under purifying selection in both species, thus, are false positives, an upper FDR bound for the snake-diverged CNEs is 11.33% (616

of 5439 CNEs). This conservatively estimates that at most 11.33% of the snake-diverged CNEs may still evolve under purifying selection in snakes.

Importantly, this set of 5439 CNEs with significant sequence divergence in snakes includes the well-studied ZRS limb enhancer that underlies *Shh* mis-expression in snakes[7,8] (Supplementary Figure 1). However, sorted by the global Z-score, the *Shh*

enhancer ranks at position 4204 (Fig. 1d), showing that numerous other CNEs, including CNEs near key limb developmental genes (see below; Supplementary Figure 2), have a more striking snake-specific divergence pattern.

**Snake-diverged CNEs are associated with limb-related genes**. If snake-diverged CNEs function as limb regulatory elements, we expect them to be preferentially located near genes with a known role in limb development. To test this, we first determined potential target genes for each CNE. Using an approach similar to GREAT[20], we associated each CNE to the closest neighboring genes that are located up to 300 Kb up- or downstream of the CNE (Supplementary Figure 3). Next, we assessed whether snake-diverged CNEs are preferentially associated with limb-related genes, compared with the remaining CNEs which are not diverged in snakes.

The analysis shows that snake-diverged CNEs are significantly enriched near genes that are involved in limb development and linked to congenital limb malformations[21] (158 of 439 genes, Fig. 2a, top panel; Supplementary Table 5). This includes *Tbx4*, *En1*, *Gli3*, *Grem1*, *Hand2*, and other genes that play important roles in limb patterning and outgrowth (see below). Furthermore, we found a significant association with genes whose knockouts in mouse result in limb phenotypes[22] (Fig. 2a, bottom panel; Supplementary Table 6). For example, 54 snake-diverged CNEs are associated with 11 genes whose knockout results in an abnormal apical ectodermal ridge, an essential signaling center required for limb outgrowth. Interestingly, the snake-diverged CNEs that are associated with limb-related genes are predominantly located outside of promoter-proximal regions (Fig. 2a, right panel), indicating divergence of distal *cis*-regulatory elements.

To confirm that these enrichments are specific to snake-diverged CNEs, we repeated the statistical tests for the 616 CNEs diverged in the limbed anole and dragon lizards, and observed no significant association with limb-related genes (Supplementary Table 6). Taken together, these results show that the limb-related enrichments are specific to CNEs diverged in snakes, indicating that mutations in these CNEs could affect expression of important limb developmental genes.

**Snake-diverged CNEs overlap limb regulatory elements**. To directly test if snake-diverged CNEs overlap *cis*-regulatory elements that are active during normal limb development in species with fully developed limbs, we performed ATAC-seq[23] in embryonic limb tissue of the tegu lizard to identify regions of accessible chromatin, which often correspond to regions with gene regulatory activity[24]. To assess tissue-specificity of limb regulatory elements, we additionally determined accessible chromatin in tegu lizard brain, heart and liver tissues, the lateral flank tissue between limb buds, and tissue of the remaining embryo. We used MACS2[25] to identify genomic regions of open chromatin (peaks) and determined tissue-specific peaks with DiffBind[26]. This analysis identified 5635 limb-specific ATAC-seq peaks (Fig. 2c; Supplementary Table 7), which are significantly associated with genes involved in limb development (GREAT analysis[20], Supplementary Figure 4) and overlap many known limb enhancers for *Fgf10*, *Grem1*, *Prrx1*, *Bmp4*, *Bmp2*, *HoxA*, *Twist1*, *Shh*, *Sox8*, and other genes (Supplementary Figure 5; Supplementary Table 9). Similarly, we identified 5417 brain-specific peaks and 6112 liver-specific peaks, which are significantly associated with genes having brain and liver functions, respectively (Supplementary Figure 4). These results validate the specificity of the ATAC-seq signal.

We found a highly significant overlap between snake-diverged CNEs and the tegu limb-specific ATAC-seq peaks (green labels in Fig. 2b), in comparison with the remaining non-diverged CNEs (Supplementary Table 8). To further corroborate this observation, we compiled a rich list of publicly available limb regulatory datasets from two other limbed species (mouse and green anole lizard), mapped the data to the tegu genome, and statistically tested the significance of the overlap (Supplementary Table 8). We found a significant overlap between snake-diverged CNEs and several limb datasets, such as limb-specific H3K27ac enhancer marks[9,27] and accessible chromatin obtained with DNase-seq[27]. Snake-diverged CNEs also significantly overlap limb-specific binding sites of the transcriptional co-activator p300[28], and binding sites of the limb-related transcription factors (TFs) HOXA13[29], HOXD13[30], PITX1[31], and GLI3[32] (Fig. 2b). Importantly, we observed the most significant overlap with a dataset of limb enhancers that was obtained by integrating many limb regulatory datasets and conserved TF-binding sites[33]. Finally, asking the question the other way around, we also found that limb regulatory elements overlap snake-diverged CNEs significantly more often than regulatory elements active in non-limb tissues (Supplementary Figure 6). In total, 933 snake-diverged CNEs, widely distributed through the FDR range, overlap at least one of the tested limb regulatory datasets (Supplementary Figure 7, Supplementary Table 4).

Notably, snake-diverged CNEs overlap many experimentally characterized limb enhancers[34] (Supplementary Tables 4 and 9), including enhancers regulating *HoxA* genes and the *HoxD* enhancers CsB and Island I[29,35,36]. In addition to the well-known *Shh* ZRS limb enhancer whose divergence likely underlies mis-expression of *Shh* in snake limb buds[7,8,37], snake-diverged CNEs overlap a VISTA limb enhancer near *Gli3*, a TF that represses *Shh* in the limb bud[38]. We also detected snake-specific sequence divergence in a VISTA limb enhancer near *Gas1*, a gene encoding a membrane protein that enhances *Shh* signaling and that is required for the normal regulatory loop between *FGF10* in the mesenchyme and *FGF8* in the apical ectodermal ridge[39] (Supplementary Figure 5B). This suggests a broader divergence in the *Shh* signaling pathway, including components that affect both expression and read-out of this important signaling morphogen.

To confirm that these enrichments are specific to limb regulatory elements, we used our ATAC-seq and other publicly available regulatory data from non-limb tissue and found no significant overlap with snake-diverged CNEs (Fig. 2b; Supplementary Table 8), with the exception of eye regulatory data (Supplementary Figure 8, Supplementary Table 10; see Discussion). In addition, we repeated the analysis using the set of CNEs diverged in the green anole and dragon lizard, and found no significant overlap with limb datasets (Supplementary Table 8), showing that the association with limb regulatory elements is specific to CNEs diverged in snakes. These results provide strong evidence that many snake-diverged CNEs overlap regulatory elements that control the expression of important limb-related genes in other limbed animals.

**Pattern of CNE divergence across the limb regulatory network**. As snake-diverged CNEs are preferentially associated with limb-related genes and overlap limb regulatory elements, we first assessed whether snake-specific regulatory divergence is limited to specific signaling pathways, or whether divergence is widespread across the limb developmental patterning network. To this end, we compiled a limb regulatory network based on previous studies[40,41] and highlighted the genes that are associated with the 933 snake-diverged CNEs that overlap limb regulatory elements (Supplementary Table 4). This analysis shows a global pattern of

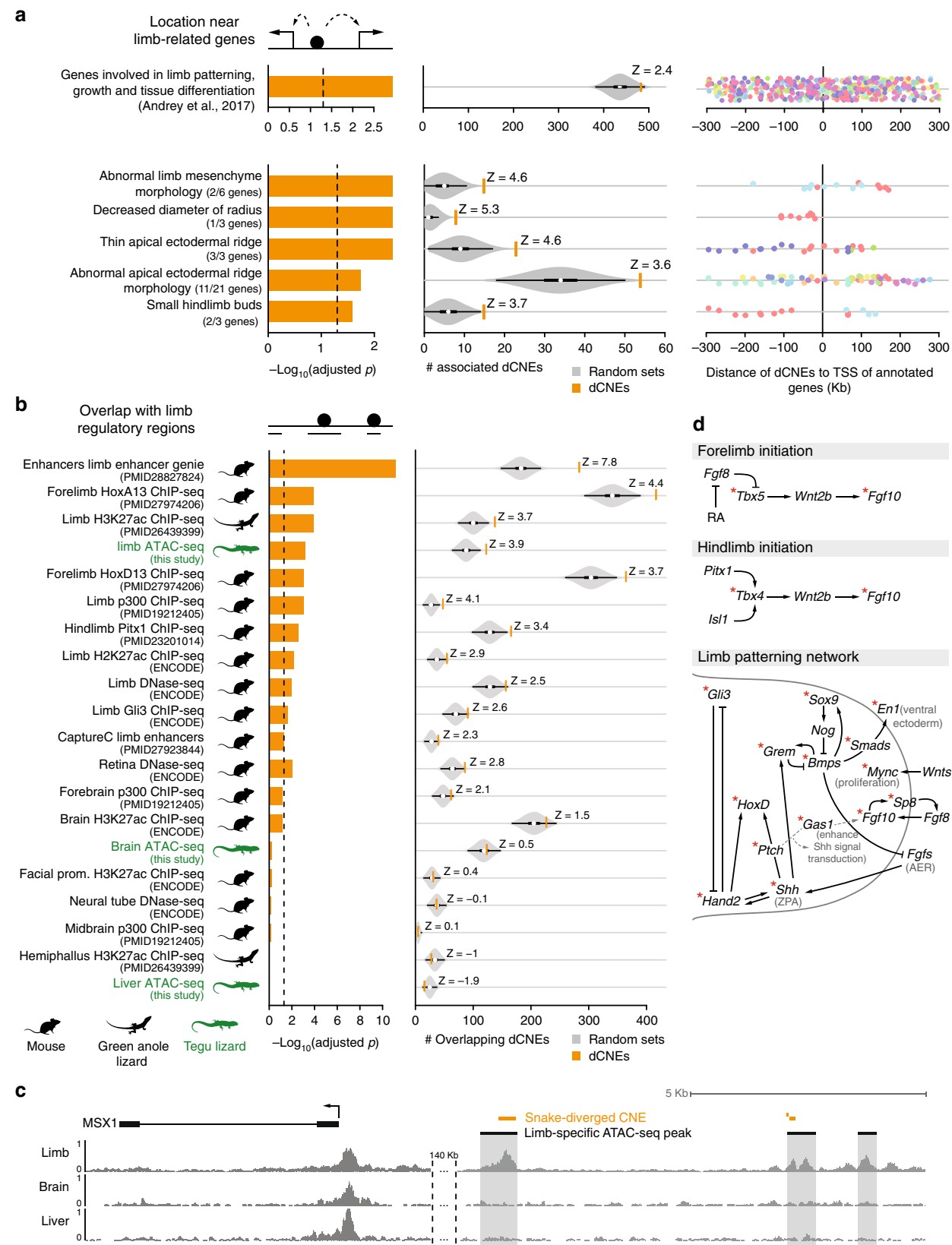

divergence affecting all the major limb signaling pathways (Fig. 2d). Consistent with widespread divergence of limb regulatory elements, we found that snake-diverged CNEs have no preferential association with genes upregulated in either fore- or hindlimbs, and no preferential overlap with fore- or hindlimb

enhancers (Supplementary Figure 9A). Notably, snake-diverged CNEs are associated with key genes controlling fore- and hindlimb initiation, anterior-posterior and dorsal-ventral patterning, and limb outgrowth, such as *Tbx4/5*, *Fgf10*, *Hand2*, *HoxD*, *Grem1*, *Shh*, *Gli3*, *Gas1*, *Sox9*, *Bmp4*, *En1*, and others. In addition

**Fig. 2** Snake-diverged CNEs are associated with limb genes and limb regulatory elements. **a** Snake-diverged CNEs are significantly associated with genes annotated with limb-related functions. Left: Bars depict Benjamini & Hochberg adjusted p-values derived by a one-sided Fisher's exact test. Middle: Observed (orange vertical bar) and expected number (gray violin plots, based on 10,000 random subsets sampled from all CNEs) of snake-diverged CNEs associated with genes in each set (the thick box inside the violin plot indicates the first quartile, the median and the third quartile). The Z-score measuring the number of standard deviations that the observed number is above the random expectation is indicated. Right panel: Many snake-diverged CNEs (dots) are far away from the transcription start site of genes in these sets. CNEs associated with the same gene have the same color. **b** Snake-diverged CNEs significantly overlap regulatory elements active in embryonic limb tissue of tegu lizard, green anole lizard, and mouse. Orange bars correspond to limb regulatory datasets. ATAC-seq datasets generated in this study are highlighted in green. Remaining visual representation as in **a**. All animal illustrations were taken from phylopic.org. **c** Genome browser screenshot shows that snake-diverged CNEs (orange) overlap tegu limb-specific ATAC-seq peaks. ATAC-seq signal tracks of limb, brain, and liver are shown. **d** Snake-diverged CNEs are broadly distributed in the limb patterning network. An asterisk marks the genes which are associated with at least one snake-diverged CNE that overlaps a limb regulatory element

to *Shh*, many of these genes cause severe limb truncations and other limb malformations when knocked-out or when their spatio-temporal expression is perturbed[7,42–48]. Furthermore, genes such as *Shh*, *Gli3*, and *Hand2* were shown to be mis-expressed in the limb buds of developing snakes[8,37]. Together, this suggests that snake-specific divergence in many limb regulatory elements may have contributed to the loss of limbs in the evolutionary history of snakes.

**Eye degeneration in subterranean mammals**. The compelling evidence for genome-wide divergence of the limb regulatory landscape in snakes raises the question whether this phenomenon is specific to limb loss or rather a general evolutionary principle. To address this question, we analyzed the fate of eye regulatory elements in four subterranean mammals in which eye degeneration evolved independently: the blind mole rat, naked mole rat, star-nosed mole, and cape golden mole. These species exhibit reduced eye sizes, disorganized lenses, thinner retinas, and loss of optic nerve connections[49–52]. A recent study has shown that a number of eye enhancers are evolving at accelerated rates in these four vision-impaired species[53]; however, the divergence of eye regulatory elements was not assessed genome-wide.

To investigate if eye degeneration is associated with genome-wide divergence of the eye regulatory landscape in the vision-impaired subterranean mammals, we conducted a comparative analysis contrasting these species with mammals that do not have a degenerated visual system. Similar to our analysis of limb loss in snakes, we created a multiple genome alignment, using mouse as the reference and including the genomes of the aforementioned four subterranean mammals, 16 other mammalian species, the green anole lizard, frog and chicken (Fig. 3a; Supplementary Table 11). With this alignment of 24 genomes, we identified 491,576 CNEs (covering 3.3% of the mouse genome), and measured sequence identity per species with respect to the reconstructed placental mammal ancestor. As the degeneration of the visual system evolved independently in the four subterranean lineages, we identified CNEs diverged in these mammals with the "Forward Genomics" branch method, that considers phylogenetic relatedness between species[18]. Of all 491,576 CNEs, this method identified 9364 CNEs with significantly higher sequence divergence in the subterranean mammals compared to the other mammals (FDR cutoff of 0.5%, Supplementary Figure 10, Supplementary Table 12).

**Diverged CNEs overlap promoters of diverged eye-related genes**. To explore if these 9364 CNEs are associated with eye-related genes, we first focused on 64 eye-specific genes that are diverged or lost in at least one of the four subterranean mammals, according to previous studies[18,54–56] (Supplementary Table 13). Since relaxed selection on these genes should result in promoter divergence, these 64 promoters should be enriched in CNEs

diverged in subterranean mammals. Indeed, 23% (12/52) of the CNEs located in the 64 promoters were identified in our genome-wide screen for divergence in subterranean mammals, which is a significant enrichment compared to 2% (357/18,033) of the CNEs located in the promoters of all other genes (Fisher's exact test: $p = 4.3e^{-10}$). Consistently, the average sequence divergence of the 52 CNEs located in these 64 promoter regions is substantially higher in the subterranean mammals (Fig. 3b). Examples illustrating promoter divergence in the lens-specific genes *Cryba4*, *Crybb1*, and *Bfsp2* are shown in Fig. 3c. Overall, CNEs located in the promoter regions of diverged eye-related genes show the expected divergence pattern in subterranean mammals.

**Diverged CNEs are associated with eye-related genes**. If divergence of the *cis*-regulatory landscape is a hallmark of lost morphological traits, we expect the CNEs diverged in subterranean mammals to be preferentially located near genes with a known role in eye development and function. Indeed, we found that diverged CNEs are significantly associated with genes whose knockout in mouse results in lens and cornea defects (Fig. 3d; Supplementary Table 14). Importantly, excluding the aforementioned 64 diverged genes results in virtually the same enrichments (Supplementary Table 15). This shows that diverged CNEs are not only associated with these diverged genes, but also with many other genes encoding morphogens or TFs that are involved in optic cup, lens, and retina development, for example *Atf4*, *Fgf9*, *Foxe3*, *Lhx2*, *Pax6*, *Prox1*, *Rax*, *Sox1*, and *Vsx2*[57]. Furthermore, most of these CNEs are far away from the transcription start site (Fig. 3d, right panel), indicating widespread divergence of distal *cis*-regulatory elements. We repeated this analysis for CNEs diverged in four mammals that do not have degenerated eyes (human, rat, cow, and elephant), and found no significant association with eye-related genes (Supplementary Table 14). Taken together, these results show that the eye-related enrichments are specific to CNEs diverged in subterranean mammals.

**Diverged CNEs overlap eye regulatory elements**. To directly test if CNEs diverged in subterranean mammals correspond to *cis*-regulatory elements active during normal eye development, we performed ATAC-seq[23] on mouse whole eye at embryonic day E11.5, and lens and retina tissues at E14.5 (Supplementary Table 1). To assess tissue-specificity of eye regulatory elements, we also determined open chromatin in mouse limb and midbrain tissues at embryonic day E11.5. This analysis resulted in 8020 eye-specific, 8401 lens-specific, and 8900 retina-specific mouse ATAC-seq peaks (Supplementary Table 16). These peaks are highly enriched near genes with eye-, lens-, and retina-related functions (Supplementary Figure 11). Furthermore, these peaks overlap characterized eye enhancers that regulate *Pax6*[58,59] and other genes[34] (Supplementary Figure 12). Likewise, 7953 limb- and 5779 midbrain-specific mouse ATAC-seq peaks are enriched

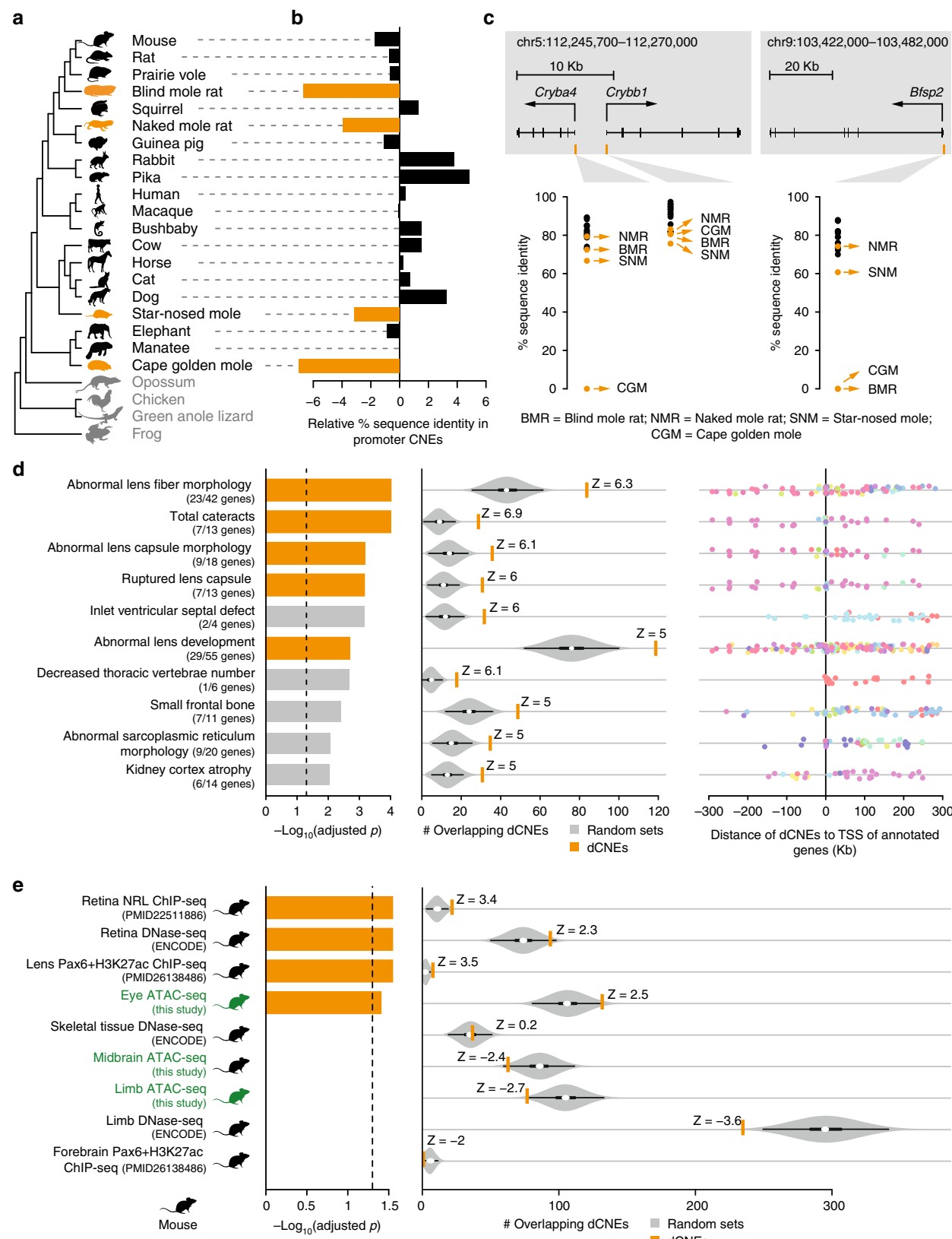

near genes with limb and brain functions, respectively, validating the specificity of the ATAC-seq signal.

We found that CNEs diverged in subterranean mammals overlap tissue-specific ATAC-seq peaks detected in the developing mouse eye, lens, and retina significantly more often than

expected from non-diverged CNEs (green labels in Fig. 3e; Supplementary Table 17). We additionally compiled a list of publicly available regulatory datasets from different eye tissues (Supplementary Table 17) and found significant overlap between diverged CNEs and several datasets, such as retina-specific

**Fig. 3** CNEs diverged in subterranean mammals are associated with eye genes and eye regulatory elements. **a** Phylogenetic tree of species in our multiple genome alignment. Subterranean mammals are in orange, mammals without degenerated eyes are in black, and outgroup species are in gray. **b** Difference in the mean percent sequence identity between 52 CNEs overlapping the promoter regions of 64 eye-specific genes, which are diverged or lost in subterranean mammals, and 18,033 CNEs overlapping promoters of other genes. Compared to other mammals, the subterranean mammals have substantially higher divergence in the CNEs overlapping the 64 eye-specific gene promoters. Promoters are defined as ±1.5 Kb around the transcription start site. **c** Three examples of diverged CNEs located in the promoter of diverged lens-specific genes. **d** Diverged CNEs are enriched near genes that lead to lens defects in a mouse knockout (top 10 enrichments are shown here; Supplementary Table 14 lists all enrichments). Lens-related knockout phenotypes are shown in orange. Left: Bars depict Benjamini & Hochberg adjusted $p$-values derived by a one-sided Fisher's exact test. Middle: Observed (orange vertical bar) and expected number (gray violin plots, based on 10,000 random subsets sampled from all CNEs) of diverged CNEs associated with phenotypes in each set (the thick box inside the violin plot indicates the first quartile, the median and the third quartile). The Z-score measuring the number of standard deviations that the observed number is above the random expectation is indicated. Right panel: Many of the diverged CNEs (dots) are far away from the transcription start site of genes in these sets. CNEs associated with the same gene have the same color. **e** Diverged CNEs significantly overlap regulatory elements active in whole eye, retina, and lens. Orange bars indicate eye regulatory datasets. ATAC-seq datasets generated in this study are highlighted in green. Remaining visual representation as in **d**. All animal illustrations were taken from phylopic.org

DNase-seq peaks[27], retina-specific binding sites of the TF NRL[60], and lens-specific enhancers bound by PAX6[61]. CNEs diverged in subterranean mammals also overlap VISTA eye enhancers (Supplementary Table 12), including an enhancer near *BMPER*, a gene that results in reduced eye size in a mouse knockout[22]. Interestingly, the CNEs diverged in subterranean mammals are preferentially associated with lens-related genes and with lens-specific ATAC-seq peaks, compared to retina-related genes and retina-specific peaks (Supplementary Figure 9B). This differential divergence signature is consistent with observations that the lenses of subterranean mammals are highly degenerated, likely because the light-focusing function of the lens became dispensable, while the reduced but normally-structured retina of these species is likely still involved in regulating the circadian rhythm[62]. In total, we found 575 diverged CNEs, widely distributed through the FDR range, that overlap at least one of the tested eye regulatory datasets (Supplementary Figure 7, Supplementary Table 12). In contrast, we found no significant overlap with our limb- and midbrain-specific ATAC-seq peaks, or other publicly available regulatory datasets from non-eye tissues (Fig. 3e, Supplementary Table 17). Additionally, CNEs diverged in human, rat, cow, and elephant showed no significant overlap with eye regulatory datasets (Supplementary Table 17).

In summary, we show that hundreds of CNEs diverged in subterranean mammals are located near genes with a role in eye development and function, and overlap regulatory elements that control gene expression during normal eye development and in adult eyes. Together with our equivalent results for snake-diverged CNEs, this corroborates that widespread divergence of the phenotype-specific *cis*-regulatory landscape is a hallmark of lost morphological traits.

**Large-scale loss of TF-binding sites**. If the sequence divergence that we consistently detected in snake-diverged CNEs and in CNEs diverged in subterranean mammals is a signature of functional decay, we expect that these diverged CNEs also lost binding sites of TFs that have a role in limb and eye tissues. On the other hand, if regulatory function is largely preserved in species exhibiting sequence divergence, we expect that mutations occurred predominantly outside of TF-binding sites, which would be conceptually similar to preserving a protein sequence in a coding region with numerous synonymous changes. To distinguish between the two possibilities, we first estimated the binding affinity of a TF to the reconstructed ancestral sequence of each CNE by computing a score based on the TF-binding preference (motif). This motif score is calculated from the summed contribution of weak and strong binding sites and makes no assumption about motif position; thus, is robust to binding site turnover. We validated this approach with available ChIP-seq

experiments, showing that the genomic regions bound by a particular TF get high motif scores for this TF (Supplementary Figure 13). Then, we selected TF-CNE combinations where a TF motif is present in the reconstructed ancestral sequence and determined the score of that particular motif in the corresponding CNE sequences of all descendant species. High motif scores imply motif conservation in the descendant species, while low scores imply motif loss. Finally, for each species, we compared the motif score distribution between diverged CNEs (foreground) and the remaining CNEs (background), considering a total of 55 (58) motifs of TFs having a role in limb (eye) development.

We first applied this strategy to analyze TF motif scores in the 933 snake-diverged CNEs that overlap limb regulatory data (Supplementary Table 4), reasoning that TFs relevant for limb development bind to these sequences. For all limbed species, we found that the median motif score of these snake-diverged CNEs is significantly higher (adjusted $p \le 7.0e^{-16}$, one-sided Wilcoxon rank sum test; Fig. 4a). This shows that limb TF motifs are significantly conserved in limbed species, which is consistent with selection acting to preserve limb regulatory function. In sharp contrast, there is a clear absence of conservation of limb TF motifs in both snakes (adjusted $p = 1$), showing that snake-specific sequence divergence results in a large-scale loss of TF-binding sites (Fig. 4a, b).

Next, we applied the same analysis to the 575 CNEs that are diverged in subterranean mammals and overlap eye regulatory data (Supplementary Table 12), reasoning that TFs relevant for eye development and function bind to these sequences. Similarly, we found that in all species without degenerated eyes, eye TF motifs are significantly conserved (adjusted $p < 1.6e^{-2}$; Fig. 4c, d), in contrast to all four subterranean mammals (adjusted $p \ge 0.13$).

It should be noted that this analysis does not imply that binding sites of limb and eye TFs are preferentially lost in snakes and subterranean mammals, respectively. Indeed, we also observed absence of conservation in TF motifs when scoring the 933 snake-diverged CNEs with eye TF motifs, and the 575 CNEs diverged in subterranean mammals with limb TF motifs (Supplementary Figures 14A-D). Furthermore, repeating this analysis with randomized TF motifs also reveals a similar binding site divergence pattern (Supplementary Figures 14E and F). This suggests that there is no selective loss of binding sites for limb or eye TFs, but rather an overall sequence divergence that affects the entire CNE. Altogether, these analyses imply that CNE divergence results in a large-scale loss of TF-binding sites, indicative of divergence of regulatory activity.

To experimentally test if sequence divergence led to change in regulatory function, we compared the regulatory activity of CNE sequences of limbed and limbless species of four snake-diverged CNEs using luciferase enhancer assays. While two CNEs do not

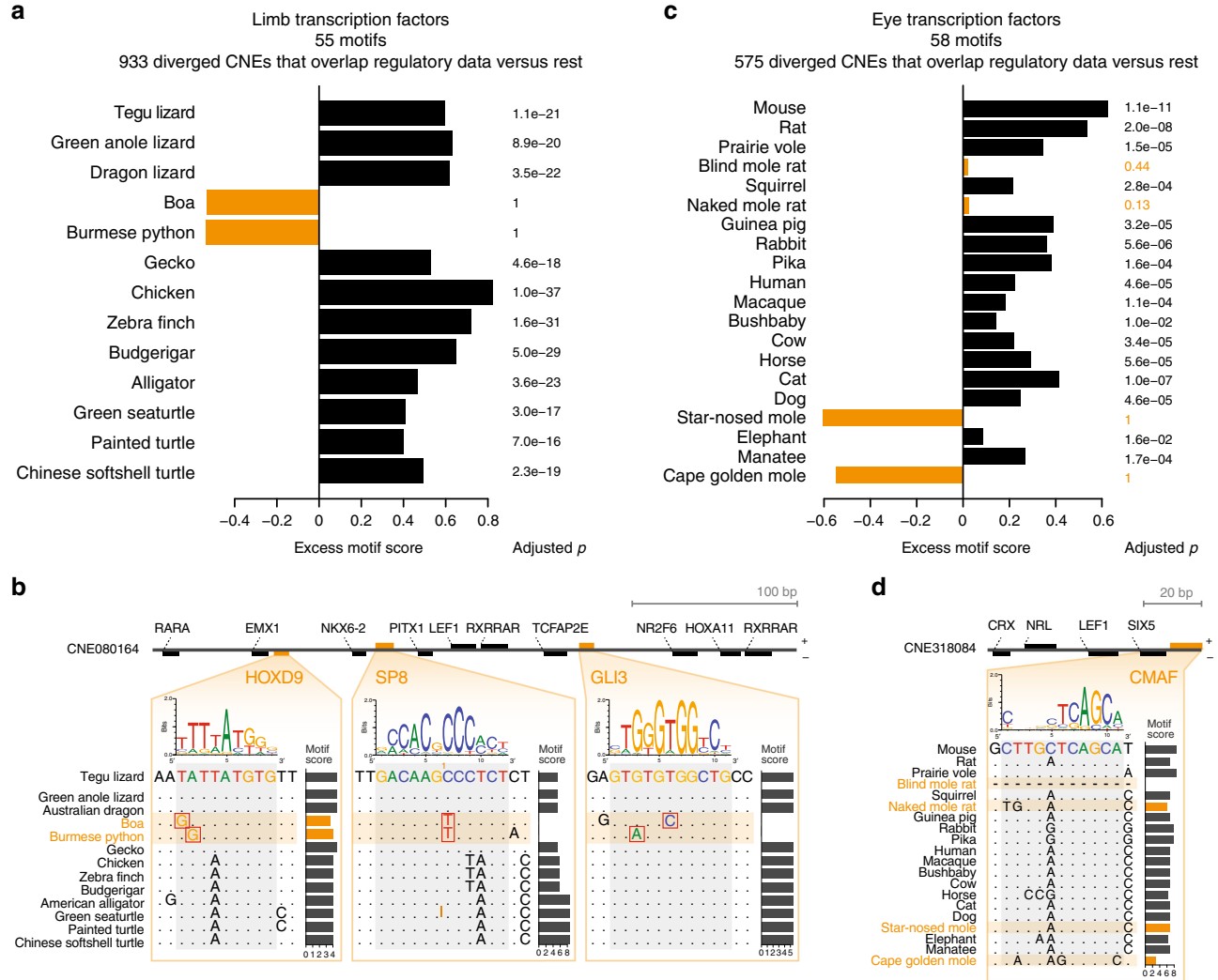

**Fig. 4** Sequence divergence results in widespread loss of transcription factor (TF) binding sites. **a** Difference in the median motif scores of limb-related TFs between 933 snake-diverged CNEs that overlap limb regulatory data and all other CNEs (excess score, *x*-axis). A positive excess score as observed for limbed species reflects a preference to preserve motifs of limb-related TFs. *P*-values comparing the distribution of motif scores of diverged and all other CNEs were computed with a one-sided Wilcoxon rank sum test and corrected for multiple testing. **b** Mutations in snakes in a CNE overlapping a limb-specific ATAC-seq peak weaken or destroy binding sites for key limb transcription factors. The binding preference of the TFs is visualized as a sequence logo, motif scores were computed with SWAN[74]. **c** As panel **a**, but using motifs of eye-related TFs to compare CNEs that are diverged in subterranean mammals and overlap eye regulatory data with all other CNEs. **d** Mutations in two subterranean mammals in a CNE overlapping a lens-specific ATAC-seq peak weaken a binding site for c-Maf, a crystallin-inducing TF that is required for lens fiber cell differentiation[75].

show enhancer activity in any species, the python sequence of the other two CNEs drives significantly different expression levels compared to the sequence of limbed species (Supplementary Figure 15). These experiments support the observation that phenotype loss is associated with the decay of the phenotype-specific *cis*-regulatory landscape.

## Discussion

Our combined comparative and functional genomics analysis shows that losses of complex morphological traits are associated with genome-wide divergence of the *cis*-regulatory landscape that is involved in the development of the trait. With regard to loss of limbs in snakes and degeneration of eyes in subterranean mammals, we consistently found that hundreds of CNEs diverged in the trait-loss species (i) are associated with genes that have key roles in development and function of this trait, (ii) significantly overlap regulatory elements that are specifically active in the respective tissues, and (iii) exhibit a clear loss signature of

relevant TF-binding sites. Together, our results provide genome-wide evidence that widespread divergence of the *cis*-regulatory landscape is a hallmark of lost complex phenotypes.

While divergence of *cis*-regulatory elements likely contributed to the loss of limbs and functional eyes, the contribution of gene divergence noticeably differs between the two traits. Eyes consist of several unique tissues and cell types, such as lens or photoreceptor cells, which express a number of non-pleiotropic genes. In contrast, genes expressed in developing and adult limbs are often pleiotropic. This difference in pleiotropy predicts a differential contribution of gene loss to the regression of limbs and eyes. Indeed, despite the fact that eye degeneration happened evolutionarily much more recent compared to limb loss, several eye-specific genes diverged and became inactivated in the subterranean mammals[18,53–56], while in snakes only the loss of *HoxD12* has been reported[63].

Whereas previous studies did not detect increased sequence divergence of many limb enhancers in snakes[8–10], we report for

the first time hundreds of limb regulatory elements that are substantially and specifically diverged in snakes. It remains an open question to what extent mutations in these regulatory elements directly contributed to the loss of limbs during evolution and to what extent sequence divergence happened as a consequence of neutral evolution after limb loss. Considering our findings that several snake-diverged limb regulatory elements are associated with genes that cause limb truncations and other malformations when perturbed, it is likely that mutations in some of these diverged regulatory elements, especially those associated with major limb patterning genes, contributed to gradual limb reduction in ancestral snakes that eventually led to complete loss of limbs. Similarly, it is plausible that mutations in some of the eye regulatory elements contributed to the gradual process of eye degeneration. In summary, our study presents a comprehensive picture of non-coding changes that may have contributed to limb loss and eye degeneration and provides many promising candidate loci to further investigate the molecular and developmental mechanisms underlying the loss of these traits.

Despite widespread divergence of limb regulatory elements in snakes, we also found many CNEs that overlap limb regulatory elements and are not substantially diverged in snakes, even after >100 My of evolution. For example, of the 5786 CNEs that overlap limb enhancers obtained by integrating several regulatory datasets and conserved TF-binding sites[33], only 315 (5.4%) have Z-scores lower than −3, corresponding to significant sequence divergence in snakes. While these 5786 limb enhancer-overlapping CNEs overall have lower Z-scores compared to the remaining CNEs (Supplementary Figure 16, two-sided Wilcoxon rank sum test $p < 2.2e\text{-}16$), indicating some degree of sequence divergence, 36% of them show no evidence for divergence in snakes (Z-score ≥ 0). One likely explanation for the maintenance of limb enhancers in limbless species is pleiotropy[10]. Notably, a previous study showed that a substantial portion of limb enhancers are also active in the developing genitals[9], implying that selection for a genital-related function may preserve many limb regulatory regions. Indeed, comparing limb regulatory elements that are conserved or diverged in snakes, we found that conserved elements have a two-fold higher tendency to overlap pleiotropic regulatory elements that are also active in genitals (Supplementary Table 18), which supports that pleiotropy is a major factor for sequence conservation. Despite being largely conserved, such pleiotropic enhancers may nevertheless lose their limb regulatory activity. This was shown for the *Tbx4* HLEB enhancer and the Prox and Island I enhancers regulating *HoxD* genes, which lost enhancer function in the developing limbs but preserve activity in the developing genitals[9,64]. While our genome-wide CNE screen detected sequence divergence in the Island I enhancer, the sequence changes that underlie such partial enhancer activity differences may often be subtle without significantly increasing overall sequence divergence (for example, the pleiotropic HLEB limb and hemiphallus enhancer[9] overlaps two CNEs with Z-scores of −1.3 and −2; Supplementary Figure 17; Supplementary Table 9). This implies that the true extent of regulatory divergence in snakes is likely higher than what we report here.

Apart from divergence signatures related to limb loss and eye degeneration, our analysis also detected signatures related to other phenotypes that changed in snakes and in vision-impaired subterranean mammals. For example, we found that snake-diverged CNEs significantly overlap regulatory elements active during normal eye development (in particular retina development; Fig. 2b, Supplementary Figure 8; Supplementary Tables 8 and 10). Together with the loss of opsins early in snake evolution[65], this enrichment is consistent with a possible subterranean origin of the crown snake lineage[66]. Furthermore, CNEs diverged

in subterranean mammals are significantly associated with genes that lead to heart beat irregularities in a mouse knockout (Supplementary Table 14). Indeed, irregularities in heart rhythmicity are known for the blind mole rat[67] and a relative of the star-nosed mole (European mole[68]). Overall, these additional enrichments indicate signatures of regulatory divergence underlying other morphological and physiological changes in snakes and subterranean mammals.

Our study suggests a general strategy to detect sequence changes in ancestral regulatory elements that potentially contribute to morphological changes. By contrasting species with and without a particular phenotype, comparative genomics screens enable the systematic identification of non-coding genomic regions that exhibit specific divergence patterns and thus may exhibit differences in regulatory activity. Leveraging functional annotations and phenotypic data from gene knockouts in model organisms, enrichment tests can reveal associations between diverged genomic regions and specific functional categories, or pathways, and thus highlight regions that are associated with promising genes. Lastly, functional genomics can further prioritize these genomic regions based on overlap with regulatory elements that are active in relevant tissues. Approaches like ATAC-seq make it now possible to also determine regulatory elements in non-model organisms, as shown here for tegu lizard embryos. Together, such a combined strategy has the potential to provide insights into the genomic basis of many morphological changes, which is fundamental to unravel the mechanisms underlying nature's phenotypic diversity.

## Methods

**Animal work**. The work with the tegu lizard was done in accordance with the Brazilian environmental and scientific legislation, under the SISBIO (Sistema de Autorização e Informação em Biodiversidade, Instituto Chico Mendes de Conservação da Biodiversidade) license 30309-4. Work with mouse was performed in accordance with the German animal welfare legislation and in strict pathogen-free conditions in the animal facility of the Max Planck Institute of Molecular Cell Biology and Genetics, Dresden, Germany. Protocols were approved by the Institutional Animal Welfare Officer (Tierschutzbeauftragter), and necessary licenses were obtained from the regional Ethical Commission for Animal Experimentation (Landesdirektion Sachsen, Dresden, Germany).

**Tegu lizard tissue sample**. To sequence the genome of the tegu lizard *Salvator merianae*, we used tissue from an adult animal collected in Mato Grosso, Brazil, and deposited on the tissue bank collection of the Zoology Department from University of São Paulo (specimen accession number LG2117).

**Whole-genome sequencing**. Genomic DNA was extracted from liver tissue of an adult animal following a standard phenol-chloroform extraction protocol. MiSeq libraries were prepared following a standard Illumina DNA library preparation, which involved shearing of DNA to 550 bp by sonication (Covaris S2), XP bead purification (Beckman Coulter), end polishing, A-tailing, and ligation of indexed adapters (TruSeq DNA PCR-Free kit, Illumina). After ligation, adapters were depleted by XP bead purification (Beckman Coulter). Libraries were quantified by qPCR with the KAPA library quantification kit (KAPA Biosystems), and equimolar pooled for sequencing. To generate mate-pair libraries, purified DNA was subjected to the Nextera Mate Pair Library Preparation protocol (Illumina). For accurate sizing after tagmentation, gel-based size selections for an average of 2 Kb and 10 Kb were done. The resulting libraries were quantified with Fragment Analyzer (Advanced Analytical) and equimolar pooled for sequencing. We sequenced 2 × 300 bp reads from three fragment libraries on the Illumina MiSeq platform, and 2 × 150 bp and 2 × 100 bp reads from two 2 Kb and two 10 Kb mate-pair libraries on the Illumina HiSeq 2500 platform to a total sequencing coverage of 104X (Supplementary Table 1).

**Genome assembly**. Raw reads were trimmed for the presence of sequencing adapters by applying cutadapt (https://github.com/marcelm/cutadapt), setting a minimum read length of 100 bp for fragment reads and 50 bp for mate-pair reads. Trimmed reads were further processed by permuting ambiguous bases using SGA (https://github.com/jts/sga; parameters '*sga preprocess --permute ambiguous*'). The resulting reads were corrected for sequencing errors applying the SGA-ICE correction pipeline (https://github.com/hillerlab/IterativeErrorCorrection), running

iterative k-mer-based corrections with increasing k-mer sizes ($k = 40$, 100, and 200 for MiSeq reads; $k = 40$, 70, 100, 125, 150 for HiSeq reads).

Error-corrected reads were assembled using ALLPATHS-LG (http://software. broadinstitute.org/allpaths-lg/blog/). We set CLOSE_UNIPATH_GAPS = False to circumvent exceedingly long runtimes related to the relatively long 300 bp fragment reads, and HAPLOIDIFY = True to account for the fact that we are assembling reads from diploid organisms. ALLPATHS-LG performs all assembly steps, including contig and scaffold assembly. The resulting 2.026 Gb assembly comprises 5988 scaffolds, with 4% of the bases located in assembly gaps.

**Repeat masking of the tegu genome.** We first applied RepeatModeler (http://www.repeatmasker.org/; parameters '-engine ncbi') to de novo identify repeat families in the tegu genome. Then, we used RepeatMasker (default parameters) with the resulting repeat library to soft-mask the tegu genome.

**Assessing assembly completeness.** We first obtained a set of non-exonic ultraconserved elements (UCEs) that are deeply conserved across vertebrates and should thus be found in a correctly assembled genome. Specifically, from all 481 UCEs (originally defined as genomic regions of at least 200 bp that are identical between human, mouse and rat[14]), we first excluded all UCEs that overlap exons in human (UCSC "ensGene" gene table). Then, we used liftOver (http://hgdownload. cse.ucsc.edu/downloads.html; parameters '-minMatch = 0.1') to obtain those UCEs that align to chicken (galGal5 assembly), zebrafish (danRer10 assembly), and medaka (oryLat2 assembly), resulting in 197 vertebrate non-exonic ultraconserved elements. We used lastz (https://www.bx.psu.edu/~rsharris/lastz/; parameters '--gappedthresh = 3000 --hspthresh = 2500 --seed = match6') to align these sequences to our tegu genome assembly and to other reptiles. All alignments were filtered for a minimum of 75% identity and a length of at least 50 bp. The number of aligning UCEs is reported in Supplementary Table 2.

Second, we ran BUSCO (https://busco.ezlab.org/) in genome mode using both the vertebrata_odb9 and tetrapoda_odb9 databases (creation date 2016-02-13 for both), which contain 2586 and 3950 highly conserved genes. The BUSCO score for the tegu lizard genome and for the genomes of other squamate reptiles are reported in Supplementary Table 2.

**Tegu transcriptome sequencing and assembly.** We extracted total RNA from two tegu lizard embryos at embryonic stage equivalent to mouse E11-E11.5. Tissues were preserved in RNAlater and total RNA was later extracted following a standard Trizol extraction. For the generation of mRNA libraries, mRNA was isolated with the NEBNext Poly(A) mRNA Magnetic Isolation Module according to the manufacturer's instructions. After chemical fragmentation, the samples were directly subjected to strand-specific RNA-Seq library preparation (Ultra Directional RNA Library Prep, NEB). For adapter ligation, custom adaptors were used (Adaptor-Oligo 1: 5′-ACA-CTC-TTT-CCC-TAC-ACG-ACG-CTC-TTC-CGA-TCT-3′, Adaptor-Oligo 2: 5′-P-GAT-CGG-AAG-AGC-ACA-CGT-CTG-AAC-TCC-AGT-CAC-3′). After ligation, unused adapters were depleted by XP bead purification (Beckman Coulter). Sample indexing was done during PCR enrichment (15 cycles). The resulting libraries were quantified with Fragment Analyzer (Advanced Analytical) and equimolar pooled for sequencing. We sequenced $2 \times 75$ bp reads from eight strand-specific mRNA libraries on the Illumina HiSeq 2500 platform (Supplementary Table 1).

We trimmed raw reads for the presence of sequencing adapters with cutadapt (https://github.com/marcelm/cutadapt), setting a minimum read length of 30 bp. Trimmed reads were mapped against the tegu genome using TopHat2 (https://ccb. jhu.edu/software/tophat/index.shtml; parameters '--library-type fr-firststrand') and assembled using Cufflinks (https://cole-trapnell-lab.github.io/cufflinks/; parameters '--library-type fr-firststrand'). In addition, we also performed a de novo transcriptome assembly with Trinity (https://github.com/trinityrnaseq/ trinityrnaseq/wiki; parameters '--SS_lib_type RF --genome_guided_max_intron 10000').

**Sequencing of open chromatin for tegu lizard and mouse.** ATAC-seq[23] was performed on 10-day old tegu lizard embryos (DH29-30; corresponding to mouse E11.5), and on E11.5 and E14.5 mouse embryos. Tegu lizard embryos were dissected in PBS to obtain samples from limb, brain, heart, liver, tissue flanking the limb buds, and the remaining embryonic material. Mouse embryos were dissected to obtain whole eye, limb, and midbrain at E11.5, and lens and retina separately at E14.5. For both tegu lizard and mouse embryos, tissue samples were pooled from at least 10 embryos and frozen in liquid nitrogen immediately after dissection. ATAC was performed on each tissue in at least two biological replicates (Supplementary Table 1). For each sample, we ground ~4 mg of tissue on a glass douncer using a milder lysis buffer containing 10% of the recommended detergent concentration. Amplification of ATAC libraries was done for 10 PCR cycles with Illumina Nextera reagents. The resulting libraries were quantified with Fragment Analyzer (Advanced Analytical) and KAPA (KAPA Biosystems) and equimolar pooled for sequencing. We sequenced $2 \times 75$ bp reads on the Illumina HiSeq 2500 and NextSeq platform (Supplementary Table 1).

We trimmed raw reads for the presence of sequencing adapters with cutadapt (https://github.com/marcelm/cutadapt), setting a minimum read length of 30 bp.

Trimmed reads were mapped against our tegu assembly or the mouse mm10 genome using Bowtie2 (http://bowtie-bio.sourceforge.net/bowtie2/index.shtml; default parameters). To center reads at the actual transposase insertion site, we shifted reads that mapped to the plus strand by $+4$ bp, and reads that mapped to the minus strand by $-5$ bp, as described in the ATAC-seq processing pipeline[23].

We used MACS2 (https://github.com/taoliu/MACS) to identify discrete peaks of enriched ATAC-seq signal in each sequencing library, using the mappable portion of the tegu genome (Hotspot getMappableSpace.pl script; https://github. com/rthurman/hotspot) and using the shifted reads as input (MACS2 parameters '--nomodel --shift -50 --extsize 100'). Tissue-specificity of ATAC-seq peaks was determined using DiffBind (https://www.bioconductor.org/packages/release/bioc/ html/DiffBind.html), providing the peak coordinates for each of the biological replicates of all tissues profiled as input, plus the mapped and shifted sequencing reads (parameters 'method = DBA_EDGER, bFullLibrarySize = FALSE, bSubControl = FALSE, bTagwise = FALSE'). All peaks identified with a $\log_2$ fold change equal or greater than 1 in one tissue compared to all others were selected as tissue-specific.

**Tegu genome annotation.** In order to annotate the tegu genome, we prepared the following four datasets:

1. Assembled tegu transcriptome: Cufflinks genome-guided transcriptome assembly (72,538 transcripts; see above) and Trinity de novo transcriptome assembly (457,763 transcripts; see above).
2. Mapping proteins of closely related species to the tegu genome: we mapped Ensembl/UniProt protein sequences from the green anole lizard *Anolis carolinensis*, and the turtles *Pelodiscus sinensis* and *Chelonia mydas* to the soft-masked tegu genome using Exonerate (https://www.ebi.ac.uk/about/ vertebrate-genomics/software/exonerate; parameters '-m protein2genome --exhaustive 0 --subopt 0 --bestn 1 --minintron 20 --maxintron 50000 --softmasktarget T --refine region --proteinhspdropoff 20').
3. Augustus gene models: To use Augustus (http://augustus.gobics.de/) for gene prediction, we first constructed a training set of gene models from the cufflinks-assembled transcripts and Trinity transcript contigs using the PASA pipeline (https://github.com/PASApipeline/PASApipeline/wiki). Trinity contigs were preprocessed with SeqClean (https://sourceforge.net/projects/ seqclean/) to remove polyA tails, vector/adapter sequences, and low-complexity regions. Next, with the PASA pipeline, we mapped the contigs to the soft-masked genome with GMAP (http://research-pub.gene.com/gmap/) and BLAT (http://hgdownload.cse.ucsc.edu/downloads.html) and removed low-quality alignments (alignment identity less than 90%, or less than 70% aligned). We then combined the remaining Trinity contig alignments with the cufflinks-assembled transcripts by collapsing redundant transcript structures and clustering overlapping transcripts. Finally, we used TransDecoder (https:// github.com/TransDecoder/TransDecoder) to search within the resulting PASA transcripts for candidate coding regions, keeping only those transcripts, which encode a complete open reading frame (with start and stop codon) of at least 100 aa. Using this set of gene models as training data, we trained Augustus on the tegu genome after hard-masking interspersed repeats.
4. CESAR gene mappings: We used CESAR[69], a Hidden Markov Model-based exon aligner that takes reading frame and splice site information into account, and our genome alignment between tegu and human (see below) to map human protein-coding genes to the tegu genome. Using 196,269 exons from 19,919 human protein-coding genes (Ensembl78, longest isoform per gene), CESAR annotated 162,959 exons with an intact reading frame, and consensus splice sites belonging to a total of 17,053 genes in the tegu lizard.

Using these four datasets, we annotated genes in the tegu genome with MAKER (http://www.yandell-lab.org/software/maker.html). Cufflinks-assembled transcripts were passed onto MAKER via the *est_gff* option on maker_opts.ctl file. The Ensembl/UniProt proteins from anole lizard and two turtles were passed via the *protein_gff* option. Augustus gene predictions were combined with CESAR mappings and were passed to MAKER via the *pred_gff* option. Furthermore, we specified the Augustus gene models in *augustus_species* in maker_opts.ctl file. The options *est2genome*, *protein2genome*, and *keep_preds* in maker_opts file were all set to 1. We ran a single iteration of MAKER, resulting in 70,462 genes and 70,486 transcripts. After filtering out all entries with an Annotation Edit Distance (AED) value of 1, the final gene annotation for the tegu genome contains 23,463 genes and 23,487 transcripts.

To obtain tegu-human orthologs, we made use of the CESAR mapping of human genes to the tegu genome as follows. First, we removed single-exon mappings shorter than 100 bp and mappings which result in very large introns (longer than 200 Kb), provided that the length of the corresponding intron in the human Ensembl transcript is greatly different (200 Kb difference). We also filtered out those MAKER gene predictions which correspond to <100 bp single-exon genes. Next, we intersected the cleaned CESAR and MAKER gene sets to obtain those tegu genes which correspond to a human gene. This resulted in a set of 16,284 tegu genes with a human ortholog.

**Multiple genome alignments.** Genome alignments were done as described in[70,71]. Briefly, to analyze limb loss in snakes, we used lastz (https://www.bx.psu.edu/

~rsharris/lastz/; parameters '$K = 2400\ L = 3000\ Y = 3000\ H = 2000$' and the HoxD55 scoring matrix) to compute pairwise local genome alignments between tegu and 28 other vertebrate genome assemblies (Supplementary Table 3). We used axtChain (http://www.soe.ucsc.edu/~kent; default parameters) to build co-linear alignment chains. For species other than the more closely related squamates (green anole and dragon lizard, boa, and python), we subsequently used more sensitive local alignments to find additional co-linear alignments by running lastz (parameters '$K = 2000\ L = 2700\ W = 5$') on all chain gaps (non-aligning regions flanked by local alignments) that are between 20 bp and 100 Kb long. All local alignments were quality-filtered by keeping only those alignments where at least one ≥30 bp region has ≥60% sequence identity and ≥1.8 bits entropy as described in ref.[70]. We used chainNet (http://www.soe.ucsc.edu/~kent; default parameters) to obtain nets from a set of chains. To remove low-scoring alignment nets that are unlikely to represent real homologies, we filtered these nets for a span in the reference and query genome of at least 4 Kb and a score of at least 20,000 (for frog and coelacanth we required a score of at least 10,000). In addition, we kept all nets that represent inversions or local translocations if they score at least 10,000. The filtered pairwise alignment nets are the input to Multiz (http://bio.cse.psu.edu/) to build a multiple alignment. The phylogenetic position of squamate species was taken from the most recent large-scale phylogenetic study of the group[72].

To analyze loss of vision in mammals, we used the same lastz/chain/net/Multiz pipeline (parameters as above) to align 20 mammals, green anole lizard, chicken, and frog (Supplementary Table 11) to the mouse mm10 assembly, which serves as the reference. We kept alignment nets with a minimum score of at least 70,000 and a minimum span of 9 Kb, or a minimum score of 150,000 and a minimum span of 6 Kb. For the three non-mammalian species, we used the additional round of sensitive local alignments, as described above, and kept alignment nets with a score of at least 15,000.

**Identification of conserved non-coding elements**. To detect evolutionarily conserved elements in the tegu and mouse multiple alignment, we used PhastCons (http://compgen.cshl.edu/phast/phastCons-HOWTO.html; parameters '*expected-length=45, target-coverage=0.3 rho=0.3*'). Since PhastCons also requires a phylogenetic tree with neutral branch lengths as input, we estimated branch lengths from 4-fold degenerated third codon positions using phyloFit from the same package (parameters '*--EM --precision HIGH --subst-mod REV*'). We combined the results of PhastCons with conserved regions detected by GERP (http://mendel.stanford.edu/SidowLab/downloads/gerp/index.html; default parameters). In addition, we required that conserved regions align well to at least 4 of the 14 mammals in the tegu-based alignment, and at least 15 of all species in the mouse-based alignment.

To obtain CNEs from the full set of conserved elements, we stringently excluded all conserved parts that overlap potentially coding regions. For the tegu genome, we obtained a comprehensive set of potentially coding sequences from four sources. First, we used proteins from the green anole lizard and the turtles *Pelodiscus sinensis* and *Chelonia mydas* that were mapped with Exonerate (see above). Second, we downloaded Refseq proteins ("vertebrate_other") from NCBI and mapped them to the tegu genome with blastx (https://blast.ncbi.nlm.nih.gov/Blast.cgi; E-value cutoff 0.01). Third and fourth, we added the full sets of CESAR human mappings and MAKER gene predictions (see above). For the well-annotated mouse genome, we obtained all coding exons listed in the UCSC tables "ensGene", "refGene", and "knownGene". A 50 bp flank was added to both sides of all coding sequences to also exclude conserved splice site regions. To obtain CNEs, we subtracted the respective coding bases from all conserved elements. Finally, we selected those CNEs that were at least 30 bp long.

**Measuring CNE sequence divergence**. To compute the percent identity between the CNE sequence of a species and the reconstructed ancestral sequence, we used the previously developed pipeline described in[18]. This pipeline uses the phylogeny-aware PRANK alignment method (https://code.google.com/p/prank-msa/; parameters '*-keep -showtree -showanc -prunetree -seed=10*') to align the sequences of the extant species and to reconstruct the most likely ancestral sequences of each CNE. It then computes the percent sequence identity between the aligned ancestral sequence and the CNE sequence of each species[17,18]. Percent sequence identity of 0 corresponds to the complete loss of CNEs[73].

**Detection of CNEs diverged in snakes**. To account for lineage-specific evolutionary rates, the resulting percent sequence identities were normalized by the total branch length between each species and the reconstructed ancestor:

$$seqId = \frac{percent\ sequence\ identity - 1}{evolutionary\ distance\ to\ ancestor}$$

Sequence identity values are thus in the negative range: the lower the sequence identity, the higher the sequence divergence, and vice versa. Both boa and python snakes belong to the same lineage, representing a single evolutionary loss. We therefore calculated a Z-score for each CNE, comparing the sequence identity value of the least diverged snake with the average and standard deviation of sequence identity values of all limbed species:

$$Z = \frac{\max(seqId_{snakes}) - avg\left(seqId_{OtherSpecies}\right)}{sd(seqId_{otherSpecies})}$$

To obtain CNEs specifically diverged in snakes, we first considered only those CNEs for which sequence identity values were calculated for both snakes, all three lizards, gecko, and at least three additional amniote species. This excludes CNEs that have missing genomic data for too many species. Second, we calculated a global Z-score by comparing snakes with all other species. Using the R package 'fdrtool' (http://strimmerlab.org/software/fdrtool/) (parameters '*statistic="normal", cutoff.method="locfdr"*'), we determined that a Z-score cutoff of −3 corresponds to an FDR of 1.186%. Thus, we removed those CNEs with a global Z-score ≥ −3. Next, we calculated a local Z-score by comparing snakes with tegu, green anole, and dragon lizards, and removed CNEs with a local Z-score ≥ −3. This resulted in 5439 snake-diverged CNEs used for further analysis.

To further estimate the FDR that corresponds to Z-score cutoffs of −3, we simulated the evolution of an ancestral genome using Evolver (http://www.drive5.com/evolver/)[17,18]. Functional genomic regions (7683 untranslated and 31,903 coding regions belonging to a total of 967 genes, 38,090 CNEs) are explicitly annotated in this genome and evolve under constraint on all branches in the real phylogenetic tree that we obtained above. As done for the real data, we used the simulated genomic data to calculate percent identity values of CNEs, normalized these values by the total branch length between each species and the amniote ancestor, and obtained global and local Z-scores for 38,090 CNEs. Since the CNEs evolve under constraint on all branches in the tree, including the branches in the snake lineage, the local and global Z-scores can be used to estimate the Z-score null distributions as any preferential sequence divergence in snakes is entirely due to chance. We found that 40 of the 38,090 simulated CNEs have global and local Z-scores < −3, which estimates the false positive rate to be 0.105%. Thus, we expect 172.7 of the 164,422 real CNEs to be false positives, which corresponds to a FDR of 3.17% (172.7 false positives in 5439 snake-diverged CNEs).

We also generated a control set of diverged CNEs using the same Z-score calculation but considering the green anole lizard and dragon lizard as trait-loss species. Using a Z-score cutoff of −3, this resulted in 616 CNEs that are more diverged in the two limbed lizards compared to all other species. These 616 CNEs were used in the enrichment analysis. To rule out that a lack of significant enrichments is due to the lower number of CNEs in this control set, we also obtained and tested a second control set consisting of the same number of CNEs (the top 5439 ranked by the global Z-score and requiring a local Z-score < −3).

We further used the set of 616 CNEs diverged in the green anole lizard and the dragon lizard to compute an upper bound of the false discovery rate. In this test, we assumed that all 616 CNEs actually evolve under purifying selection in both lizards, which is likely not the case as these CNEs could be related to phenotypic changes that are shared between these two species. Nevertheless, making this assumption, one would expect that 616 of the 5439 snake-diverged CNEs are false positives, which corresponds to an upper bound of the FDR of 11.33%.

**Detection of CNEs diverged in subterranean mammals**. In contrast to snakes, the degeneration of the visual system has happened independently in four subterranean mammals. Therefore, we applied the branch method of the "Forward Genomics" approach[18] to select CNEs that are significantly more diverged in the four vision-impaired species compared to all other species. The branch method considers both phylogenetic relatedness between species and differences in their evolutionary rates, and computes the significance of the association between the sequence divergence and phenotype. CNEs for which sequence identity values could not be calculated for all four vision-impaired species were excluded. The resulting p-values were corrected for multiple testing using the Benjamini & Hochberg method. Using an adjusted p-value cutoff of 0.005, this resulted in 9364 diverged CNEs used for further analysis.

We generated a control set of diverged CNEs using the same Forward Genomics setup but considering human, rat, cow, and elephant as trait-loss species. Using the same adjusted p-value cutoff, this resulted in 86 CNEs that are more diverged in these four mammals compared to all other species. As for snake-diverged CNEs, we also obtained a second control set with the same number of CNEs by taking the top-ranked 9364 CNEs.

**Functional enrichment analysis**. We statistically tested if diverged CNEs are significantly associated with genes belonging to a functional class, using all remaining non-diverged CNEs as a control. To associate CNEs to potential target genes, we used the regulatory domain concept of GREAT (http://great.stanford.edu/). Briefly, for each gene, we defined a basal (promoter-associated) domain of 5 Kb upstream and 1 Kb downstream of the transcription start site and a distal domain that extends the basal domain up to the basal domain of the next gene or at most 300 Kb in either direction (Supplementary Figure 3A). Transcription start sites for the mouse mm10 genome were downloaded from the GREAT website. The overlap of a CNE with one or more regulatory domains is then used to associate the CNE to its target gene(s).

To obtain functional annotations for all genes genome-wide, we used the Mouse Genome Informatics (MGI) Phenotype ontology (http://www.informatics.jax.org) that lists phenotypes observed in mouse gene knockouts (MGI table MGI_PhenoGenoMP.rpt). We only considered phenotypes resulting from single gene knockouts. All phenotypes associated to at least three genes were tested for enrichment. For the snake-diverged CNE analysis, we mapped MGI phenotypes to the corresponding tegu genes by using our CESAR annotation (see above). We first tested enrichments of snake-diverged CNEs using all mouse knockout phenotypes. However, this analysis was not meaningful as the top enriched terms were biased towards mammalian-specific characteristics such as placenta, milk, brown fat, or testicular descent, which reptiles never possessed. Therefore, we specifically tested enrichment of limb MGI knockout phenotypes (all child terms of MP:0002109 "abnormal limb morphology"). In addition to MGI annotations, we used a set of 435 genes that are involved in limb patterning, growth, and tissue differentiation[21]. For the analysis of CNEs diverged in vision-impaired subterranean mammals, we tested all MGI knockout phenotypes.

For each gene set (genes with the same functional annotation), we used the LOLA R package (https://www.bioconductor.org/packages/release/bioc/html/LOLA.html), which implements a one-sided Fisher's exact test, to ask whether diverged CNEs overlap the regulatory domains of this gene set more often than the remaining non-diverged CNEs. To exclude the possibility that significant results are explained by several CNEs that are in close proximity, we merged all CNEs closer than 50 bp into one CNE for these tests, reducing the number of diverged CNEs to 4849 for the limb study and 9259 for the eye study. The resulting p-values were corrected for multiple testing using the Benjamini & Hochberg method. To further estimate the magnitude of an enrichment as a Z-score, we randomly subsampled, 10,000 times, same-sized sets from the full set of CNEs and determined the overlap with regulatory domains of genes in this set.

To test if diverged CNEs significantly overlap regulatory elements active in a particular tissue, we used LOLA and random subsampling as we did for regulatory domains of gene sets (see above). In addition to our ATAC-seq peaks, we obtained a large number of publicly available regulatory datasets (see Supplementary Tables 8 and 17 for references) to get tissue-specific peaks. In short, for each experimental assay, we selected peaks observed in at least two tissue/stage replicates (when available), and obtained tissue-specific peaks by subtracting peaks observed with this assay in other tissues (when available). To test for enrichments of snake-diverged CNEs, coordinates of regulatory elements were mapped to the tegu genome (liftOver, parameters '-minMatch=0.1'). Details about all limb and eye datasets, the underlying experiments, tissues, time points, and species, and the overlap statistics used for LOLA's one-sided Fisher's exact test are provided in Supplementary Tables 8 and 17.

**Scoring transcription factor binding sites**. A motif library was generated by integrating motifs from TRANSFAC (http://gene-regulation.com/pub/databases.html), JASPAR (http://jaspar.genereg.net/), and UniPROBE (http://the_brain.bwh.harvard.edu/uniprobe/). We clustered the resulting 2272 motifs into 638 groups to reduce motif redundancy based on a similarity score computed with TomTom (http://meme-suite.org/tools/tomtom; parameters '-thresh=1 -dist=ed') (Supplementary Table 19). We then scored each CNE sequence of each species for the presence of each of the motifs by applying SWAN (http://europa.cs.uiuc.edu/CompGenomics09/; parameters 'window length 200 bp; GC-matched random generated background sequence; -wc'), a Hidden Markov Model-based method that considers both weak and strong motif occurrences and avoids inflated scores in case motif occurrences overlap each other. Since SWAN scores do not consider the exact position of each motif and since we consider the sequence of each species separately, our results are not affected by turnover of the same motif in the CNE sequence.

To determine if sequence divergence resulted in the loss of well-conserved TF motifs, we first selected those CNE-motif pairs for which the reconstructed ancestral sequence had a score exceeding random expectation. Since different TFs have different binding strengths, we calculated a stringent TF-specific score cutoff that indicates the presence of at least one TF motif in an otherwise random sequence. To this end, we generated 100 random sequences of length 200 bp each for 21 GC-content bins that range from 0 to 100%. Then, we inserted a binding site (sampled from the motif) into the random sequences, applied SWAN, and determined the 90% score quantile of each bin. This 90% score quantile corresponds to a GC-dependent motif cutoff, where 10% of the GC-matched random sequences with the inserted motif have a score at least as high. Then, we only considered CNE-motif pairs, for which the reconstructed ancestral sequence scored above this threshold. For those CNEs, we then determined the score of this motif for the CNE sequence of each species.

For the limb study, we compiled a list of 65 TFs that are known to function in limb development and created a library of 55 motifs for these TFs (Supplementary Table 20). CNEs were separated into two groups: (i) snake-diverged CNEs that overlap limb-specific regulatory data (Supplementary Table 4), and (ii) all other CNEs. A one-sided Wilcoxon rank sum test was used to assess if diverged CNEs have significantly lower motif scores than the other CNEs. Resulting p-values were corrected for multiple testing using the Benjamini & Hochberg method. We also computed the median score for CNE-motif pairs separately for both CNE groups and visualized the difference in the median in Fig. 4a. Accordingly, for the eye study, we compiled a list of 60 TFs that are known to function in eye development,

resulting in 58 motifs contained in our library (Supplementary Table 21) and visualized the difference in the median between CNEs diverged in subterranean mammals that overlap eye-specific regulatory data (Supplementary Table 12) and all other CNEs in Fig. 4c.

To obtain limb TF motifs that have a different binding motif compared to eye TFs (and vice versa), we computed pairwise similarity scores with TomTom (http://meme-suite.org/tools/tomtom; parameters '-thresh 1 -dist ed') and removed motifs of limb TFs that have a pairwise similarity score ≤ 0.01 to another motif of an eye TF. To obtain randomized limb and eye TF motifs, we implemented a column-wise replacement procedure: starting from our motif library, we sorted all 7567 motif columns according to their information content into 20 bins. Each limb or eye TF motif was then resampled by replacing every column by a randomly chosen column from the corresponding information content bin.

**Luciferase assays**. We performed a dual luciferase assay to test diverged CNEs for regulatory activity. We synthesized the CNE sequences of mouse, anolis/tegu, and python of five different snake-diverged CNEs (including the CNE which overlaps the *Shh* ZRS limb enhancer), which were cloned into a pGL4.23[luc2/minP] vector (Promega) (Supplementary Table 22). The ZRS was used as a positive control for the assay, and the four other CNEs were selected based on their overlap with limb regulatory data and proximity to genes known to be involved in limb development.

NIH-3T3 cells (DSMZ) were cultured in DMEM medium (Thermo Fisher Scientific) supplemented with 10% fetal calf serum (Sigma), at 37 °C, 5% CO$_2$, and 100% humidity. For detachment, the cells were washed with PBS and treated with 0.05% Trypsin/EDTA (Thermo Fisher Scientific) for 3 min at 37 °C. After incubation fresh medium was added, cells were spun down at 140× *g* for 5 min and re-suspended in fresh complete medium solution. 2500 NIH-3T3 cells/well were seeded into 384-well plates (Corning).

Enhancer constructs were transfected 24 h after seeding using FuGENE6 (Promega) as the transfection reagent, following manufacturer's instructions. In brief, FuGENE6 was mixed in OptiMEM (Thermo Fisher Scientific) and incubated for 5 min at room temperature. The plasmids with firefly (pGL4.23[luc2/minP]; Promega) and renilla (pGL4.73[hRluc/SV40]; Promega) luciferases were added to the mix in a ratio of 100:1 and, after an incubation period of 20 min at room temperature, the transfection complex was added to the cells. The optimal FuGENE/DNA ratio for NIH-3T3 cells was set to 8:1.

The read-out of the luminescence was done 24 h after transfection, using the Dual-Glo Luciferase Assay System Kit (Promega) as substrate, according to manufacturer's instructions. For the read-out of the luminescence signal we used an Envision 2104 Multilabel reader (Perkin Elmer) with an ultra-sensitive luminescence 384-well aperture.

The assay for each CNE was repeated five times (five different plates), with a total of six technical replicates per plate for each CNE of each of the tested species. The ratio of firefly and renilla luciferases was normalized by the mean of the replicates of the empty control vector of the respective plate. Significance was assessed using a two-sided Wilcoxon rank sum test.

**Code Availability**. Our study used publicly available software. All tools and methods, references, version number, and the used parameters are detailed in Supplementary Table 23.

## Data availability

All primary data, including the tegu lizard genome assembly, its gene annotation, CNEs, ATAC-seq peaks, the tegu multiple genome alignment, and the mouse multiple genome alignment are available at https://bds.mpi-cbg.de/hillerlab/CNEDivergence/. All Illumina sequencing data and the tegu genome assembly has been deposited to NCBI under the project accession numbers PRJNA473319, PRJNA481646, and PRJNA481520.

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

## Acknowledgements

We thank the genomics community for sequencing and assembling the genomes of the many vertebrates used here, and the UCSC genome browser group for providing software and genome annotations. We also thank Virag Sharma for running CESAR, Nadine Vastenhouw for experimental infrastructure, Tiago Ferreira for helping with dissection of mouse eyes, Terence Capellini for helpful comments on the manuscript and on the experimental assay, Aliona Bogdanova for cloning of plasmids, Jared Simpson for help with SGA, and Andreas Dahl and Sylke Winkler for DNA sequencing, the Computer Service Facilities of the MPI-CBG and MPI-PKS, and the DNA Sequencing, Microarray and Biomedical service facilities of the MPI-CBG for their support. This work was supported by the Max Planck Society, and by FAPESP stipends 2012/01319-8 and 2012/23360 to J.G.R.

## Author contributions

J.G.R. annotated the tegu genome, performed ATAC-seq, and analyzed the limb and eye loss trait. K.S. assembled and annotated the tegu genome and analyzed the limb and eye loss trait. G.P. performed the initial eye loss analysis. B.E.L. performed the TF-binding site analysis. A.P. helped with genome annotation. C.M., M.B. and J.G.R. performed luciferase assays. M.T.R. provided samples. M.H. conceived and supervised the study and performed data analysis. M.H., J.G.R. and K.S. wrote the manuscript. K.S. and J.G.R. made figures. All authors approved the final manuscript.

## Additional information

**Competing interests:** The authors declare no competing interests

