## [Peer Review File · Nature Communications]

Reviewers' comments:

Reviewer #1 (Remarks to the Author):

In their manuscript, the authors present the case for widespread divergence in conserved noncoding regions associated with lost traits. In doing so, they present the new genome sequence of the tegu lizard and new ATAC-seq results. Their analysis centers on measuring the normalized divergence in limbless reptiles (snakes) as compared to other species in their alignments. When compounded over thousands of conserved non-coding, they analyze the most divergent tail of that distribution. These regions are significantly enriched near genes affecting limb phenotypes and near ATAC-seq peaks more specific to limb.

The authors perform a different type of analysis, although with a broadly similar strategy, on loss of eyesight in subterranean mammals. The most divergent regions are again statistically closer to genes affecting eye phenotypes.

The study is largely technically sound, and could represent a nice contribution to the recent surge in comparative genomics. Below, we present a number of scientific and technical concerns that should be addressed.

Major Concerns:

1) The divergent regions associated with trait loss are shown in general to lie near regions that also have evidence for affecting the phenotype in question, but the strength of that enrichment is not well defined in their report. How reliable is an individual inference of divergence? That is, for a single region? It is unclear what the false discovery rate would be.

The authors must provide some kind of measure or prediction of the accuracy of an single inference in their divergent regions. This is especially important if they wish these regions to be of use for future developmental biology experiments, as alluded to in Discussion.

2) Similarly, to strengthen the premise of the paper, the authors should determine an estimated number of CNEs that have diverged across the genome. This would require some assumptions about power and regions analyzed, but would be an important guideline for the number of regions specifically affecting limb development.

3) The limbless analysis uses a normalization of divergence and then a Z-score cutoff. Is the seqID statistic normally distributed? That is an assumption of using the Z-score. It's not clear whether this strategy is statistically sound.

4) Rather than choosing an arbitrary cutoff of $Z < -3$, the authors should apply some type of false discovery rate correction or alternatively examine the tail of the distribution and measure enrichment along it. This opportunity is missed.

5) The manuscript does not sufficiently explain how regions were scored as an ATAC-seq peak, and how tissue-specificity of a peak was inferred.

6) Pg 6. Snake-diverged CNEs significantly overlap limb regulatory elements. 933 out of the total 5000 odd CNEs in total show overlap with any limb regulatory. Where do these 933 regions rank in terms of sequence divergence? It will be interesting to see if they are near the top or span the entire range, considering the ZRS enhancer ranks 4205 among the diverged CNEs

7) Pg 18. Detection of CNEs diverged in snakes: The boa and python snake lineage is considered to represent a single evolutionary loss event. Since their divergence from the MRCA, the authors argue

the sequences are neutrally evolving. Given that, why do they not use them as two loss events? This would also mean they can use their Forward Genomics method to detect CNEs associated with the trait, meaning they have a consistent method across the two traits they study (limb and eye degeneration).

8) Pg10. CNE divergence in TF binding sites: To understand if the diverged CNEs are associated with lost TFBS, the authors compare the motif score distribution for diverged CNEs against background CNEs. But the choice of CNEs are already informed by regulatory data from ATAC-seq and other experiments as they consider only CNEs that they detect as overlapping with regulatory data. This is a biased set, and perhaps a better test would be to perform the same analysis with all the CNEs they detect as being diverged.

Minor Concern:

9) Page 7. It is awkward to have the citations in superscript in the middle of the sentence. "compiled a limb regulatory network based on 60-63 and..." Perhaps "...based on previous studies..." and then cite at the end of the sentence.

Reviewer #2 (Remarks to the Author):

The great morphological diversity between species has sparked much of biologists' interest into their underlying genetic mechanisms. Both protein-coding sequences, and their regulatory elements which control genes' spatiotemporal expression can account for the morphological changes. However, it has been proposed that due to the pleiotropic effect of gene sequences, cis-regulatory elements might have a greater contribution and have become recent focus of research. This paper selected two generally very conserved traits: limb and eye, which have been lost independently several times throughout vertebrate evolution. They identified 5,439 conserved non-coding elements (CNEs) that are specifically diverged in the snake, with 933 overlapped with putative limb regulatory datasets. These snake diverged CNEs are also significantly associated with known limb-patterning genes, supporting their important role in limb degeneration of snakes. Similarly, they identified 9,364 CNEs diverged in the subterranean mammals (mole rat species etc.) but conserved in other species, with 575 overlapped with tested eye regulatory datasets. Both together provide evidence for the divergence of cis-regulatory elements participating in the lost of certain complex traits. I think the genomic datasets generated from this study is very valuable, including one additional lizard's genome, Tegu lizard limb ATAC-seq, mouse eye ATAC-seq. However, the new insights that are offered from these datasets are not substantial. As it is expected that eye or limb related cis-regulatory elements would become diverged in species that have lost these traits. However, given there are even much larger numbers of other diverged CNEs (>80% of snake diverged CNEs, and >90% of subterranean mammals CNEs) that are not related to these traits, one may directly ask the potential reasons for such a high false positive rate. I have other detailed comments below:

1. I recognize that it is very difficult to directly identify limb enhancers and their targeting genes. The authors basically ask in the paper: how many snake-specific diverged CNEs are correlated with limb enhancers or limb patterning genes? I am curious about the question being asked the other way around, how many known limb enhancers (e.g., Monti et al. 2017) are overlapped with these snake diverged CNEs? As besides limb, many other traits may have undergone degenerative evolution (e.g., vision) in snakes, which may directly account for the rest CNEs that do not overlap with limb enhancers/genes.
2. In snakes, it is known from fossil data that forelimbs have become lost before the hindlimbs. Do the snake diverged CNEs or limb-ATAC peaks have a different distribution regarding their overlaps with forelimb and hindlimb genes/enhancers?
3. The putative mouse limb enhancers and their targeted genes have been recently characterized by

Capture-C (Andrey et al. 2017). Among those ~400 limb-patterning genes and their inferred regulatory elements, how many are overlapped with the snake diverged CNEs?

4. Although the patterns of eye and limb loss regarding the cis-regulatory elements are similar, their experienced time-scale are different: the eye loss is much more recent than the limb loss of snakes. I wonder if it is possible to test using the current datasets that protein coding genes and cis-regulatory elements may have different contributions at different evolution stage. Or at least this should be discussed, otherwise, the two datasets and the conclusion look redundant in the paper.

5. Other minor points:

1) Page 3, I think here the previous studies have demonstrated that the retention of many limb-related enhancers is due to these enhancers' pleiotropy—that they are related to genital development as well. This should be mentioned in the introduction. As not only genes, but also regulatory elements can be also pleiotropic.

2) Page 4, it is recommended to use BUSCO gene set to test the completeness of a draft genome. As the currently used ultraconserved elements were published long time ago, maybe either incomplete or not sufficient for the whole-genome coverage examination.

3) Page 5, many other factors, like mutation coldspot or genetic drift could also lead to the formation of CNEs.

Reviewer #3 (Remarks to the Author):

A major goal in evolutionary biology is identifying specific non-coding DNAs (including enhancers) responsible for the morphological change. In this paper, Hiller and colleagues use comparative genomics to determine a candidate set of enhancers that are associated with limb and vision loss in snakes and subterranean mammals respectively. First, the authors use multiple sequence alignment between many vertebrate genomes to identify conserved non-coding elements (CNE) that are specifically diverged in snakes and subterranean mammals. The authors then use various published gene expression and enhancer epigenomics data to show that indeed many of these diverged CNEs are enriched around limb/eye genes and overlap candidate limb/eye enhancers. Finally, they show that relevant TF binding sites are lost from diverged CNE, providing a potential mechanism for CNE deactivation during evolution.

The finding that limb and eye enhancers diverged/lost function in snakes and subterranean mammals has in principle been described by other groups before (e.g., (Infante et al. 2015; Partha et al. 2017)), so it is not unexpected per se. However, the value of the present work is that it provides an unbiased, purely sequence-based analysis framework for the identification of such divergent enhancers genome-wide, which should have wide applications for other phenotypes/species with a similar phenotype loss. The computational strategy itself is not particularly novel but represents a significant extension of a similar method by the same author (Hiller et al., NAR and Cell Rep, 2012). The new method focuses more on sequence divergence rather than a complete loss of non-coding DNA sequence, which is important given that enhancers often lose their function despite clearly recognizable sequence conservation (e.g., (Infante et al. 2015; Leal and Cohn 2016; Kvon et al. 2016)). Genome-wide list of candidate diverged CNEs will be a valuable resource for evolutionary biologists studying limb loss in snakes and vision loss in subterranean mammals. The authors also generated significant new experimental data (Tegu lizard whole genome sequencing and ATAC-seq data from lizard tissues) which, beyond their immediate application in the present manuscript, will be valuable resources for the scientific community. The paper is for the most part well written, although in some places it is fragmented and hard to follow - mainly because it is separated into independent limb and vision loss parts. This can be improved. Figures are very well designed and easy to understand.

While the presented analysis is rigorous and conclusions are sound, the paper could be significantly strengthened by addition of at least some experimental data empirically validating some of the findings. Specifically, it would be reassuring to see at least a few examples of candidate CNE that indeed lost/diverged their activity in an experimental setup to rule out the possibility that observed sequence changes are neutral with respect to CNE activity.

Major comments:

1) Empirical validation for diverged CNEs. It is nice to see that diverged CNE overlap putative limb/eye enhancers and are associated with limb/eye gene programs, especially because their identification was purely based on the analysis of sequence alignments and is independent of gene expression and epigenomics data. However, one would like to see functional validation of the CNE divergence in a model organism, for example using transgenic reporter assays. This would strongly support the main conclusion of the paper and rule out a possibility that these sequence changes are neutral with respect to CNE activity.

2) Related to the previous comment. The authors show that their set of diverged CNEs includes ZRS limb enhancers which were shown to be degraded in the snake lineage. What about other known limb enhancers that diverged in snakes (e.g., HLEA and HLEB (Infante et al. 2015))? Was their algorithm able to find them?

3) Page 10, loss of TFBS. Is the specific loss of TF motifs in diverged CNE simply due to their overall poor conservation in snakes/subterranean mammals? An appropriate control would be comparing motif matches for limb TF motifs in eye enhancers and vice versa, for eye TF motifs in limb enhancers. If they also show decay, the motif loss is likely due to overall CNE sequence divergence.

4) The authors speculate in the discussion about limb reduction in snakes having occurred as a stepwise process, with gradual reduction of limbs. The data set presented here may offer the intriguing possibility to directly see the evolutionary signature of this gradual process. The substitution rate in individual degrading CNEs in snakes should in principle be directly correlated to the time from loss of function of that CNE. With suitable corrections for the evolutionary constraint of a given sequence in limbed vertebrates, the present data set should in principle offer the opportunity to determine the age of the loss of function in individual CNEs. That said, due to the relatively short length and limited number of substitutions in each CNE, this estimate will likely not be very accurate at the level of individual CNEs. However, it should be possible to model the expected distribution of substitution rates across the entire CNE set assuming an instantaneous loss of constraint on all limb enhancers at the root of the snakes. If the observed distribution is shifted compared to this model, i.e. if there are more than expected lower-substitution CNEs, this could be a direct indication of a gradual process. Depending on signal strength, it may even be possible to observe bi-/multimodal distributions corresponding to groups of enhancers associated with specific morphological features. If the data set is underpowered for this type of analysis, another possibility would be to ask if CNEs associated with genes responsible for basic/early processes (e.g. limb bud induction/differentiation) show an overall lower substitution rate than those involved in later limb developmental processes (bone elongation etc.) – if it was a gradual process, the enhancers should lose function in “reverse order” of the developmental events.

5) The section on page 12, starting with “Interestingly, our analysis...” was slightly confusing as it attempts to link the two phenotypes in this study, which up to this point are presented and analyzed independently. The way this is currently presented, the fact that snakes also have poor vision and have been speculated to have subterranean ancestors appears like a post-hoc explanation of

unexpected observations. Overall this entire paragraph seems speculative and may not be needed. Alternatively, if loss of vision and/or underground lifestyle in snake ancestors seems important, it should be introduced earlier and considered in the Results section. But I don't think this would help with the narrative, I view this as a peripheral observation that distracts from an otherwise clean study design and analysis strategy focused on the core question of this work.

Minor comments:

1) Abstract, last sentence: "Together, our results provide the first evidence that genome-wide decay of the phenotype-specific cis-regulatory landscape is a hallmark of lost morphological traits." - The work does not represent the "first evidence." See above papers (especially Partha et al. 2017) showing the same trend for eye enhancers in mole rats and moles. Please tone down this sentence.

2) Please include line numbers in the next submission.

References:

Infante CR, Mihala AG, Park S, Wang JS, Johnson KK, Lauderdale JD, Menke DB. 2015. Shared Enhancer Activity in the Limbs and Phallus and Functional Divergence of a Limb-Genital cis-Regulatory Element in Snakes. *Developmental Cell* 1–32.

Kvon EZ, Kamneva OK, Melo US, Barozzi I, Osterwalder M, Mannion BJ, Tissières V, Pickle CS, Plajzer-Frick I, Lee EA, et al. 2016. Progressive Loss of Function in a Limb Enhancer during Snake Evolution. *Cell* 167: 633–642.e11.

Leal F, Cohn MJ. 2016. Loss and Re-emergence of Legs in Snakes by Modular Evolution of Sonic hedgehog and HOXD Enhancers. *Curr Biol*.

Partha R, Chauhan BK, Ferreira Z, Robinson JD, Lathrop K, Nischal KK, Chikina M, Clark NL. 2017. Subterranean mammals show convergent regression in ocular genes and enhancers, along with adaptation to tunneling. *Elife* 6: 1519.

Reviewer #1 (Remarks to the Author):

In their manuscript, the authors present the case for widespread divergence in conserved noncoding regions associated with lost traits. In doing so, they present the new genome sequence of the tegu lizard and new ATAC-seq results. Their analysis centers on measuring the normalized divergence in limbless reptiles (snakes) as compared to other species in their alignments. When compounded over thousands of conserved non-coding, they analyze the most divergent tail of that distribution. These regions are significantly enriched near genes affecting limb phenotypes and near ATAC-seq peaks more specific to limb.

The authors perform a different type of analysis, although with a broadly similar strategy, on loss of eyesight in subterranean mammals. The most divergent regions are again statistically closer to genes affecting eye phenotypes.

The study is largely technically sound, and could represent a nice contribution to the recent surge in comparative genomics. Below, we present a number of scientific and technical concerns that should be addressed.

Major Concerns:

1) The divergent regions associated with trait loss are shown in general to lie near regions that also have evidence for affecting the phenotype in question, but the strength of that enrichment is not well defined in their report. How reliable is an individual inference of divergence? That is, for a single region? It is unclear what the false discovery rate would be. The authors must provide some kind of measure or prediction of the accuracy of an single inference in their divergent regions. This is especially important if they wish these regions to be of use for future developmental biology experiments, as alluded to in Discussion.

To measure sequence divergence of a single region, we reconstructed ancestral states with PRANK, a Maximum Likelihood approach, on the known species phylogeny. PRANK is widely used and has been reported to produce the most accurate alignments compared to other methods (Fletcher & Yang, MBE, 2010; Jordan & Goldman, MBE, 2012).

We thank the reviewer for bringing up the issue of estimating false discovery rates. For the CNEs preferentially diverged in subterranean mammals, we used an adjusted p -value cutoff of 0.5%, as described on page 9 (for clarity, we have replaced this phrase now with "FDR cutoff of 0.5%"). The reviewer is correct that we did not compute an FDR for snake-diverged CNEs. Therefore, we have now used the `fdrtool` package (Strimmer, BMC Bioinformatics, 2008) to calculate an FDR rate for the Z-scores of the CNEs, which shows that the false discovery rate that corresponds to our Z-score cutoff of -3 is 1.186%. This indicates that our set of 5,439 snake-diverged CNEs contains a rather small percentage of false positives. We included this analysis in the results and methods sections:

Pg.21, methods:

"Using the R package 'fdrtool'¹³⁰ (parameters 'statistic="normal", cutoff.method="locfdr"), we determined that a Z-score cutoff of -3 corresponds to an FDR of 1.186%."

Pg.5, results:

"Requiring a Z-score cutoff of -3 for both comparisons (false discovery rate FDR of 1,19%; Supplementary Table 4), we identified 5,439 CNEs that are highly and specifically diverged in snakes"

We also included both the Z-scores and FDR values for each snake-diverged CNE in Supplementary table 4.

To test if the diverged CNEs are significantly associated with genes belonging to a functional group or regulatory elements active in certain tissues, we used a one-sided Fisher's exact test (because we are only interested in enrichments, not depletions) implemented in the LOLA R package, and controlled for multiple testing. This is described in the methods, page 23, and illustrated in Supplementary Figure 3. The corrected p -values are shown in Figures 2 and 3.

Like the reviewer, we were also concerned about the "strength of that enrichment", as significant p -values can also arise from large sample sizes with minute differences. Therefore, we estimated the strength of the enrichment by computing a Z-score using 10,000 randomly chosen same-sized sets from the entire CNE set. This analysis resulted in Z-scores that are always greater than 2, and often much higher, as shown in the middle panel of Figures 2A, 2B and 3D, 3E.

2) Similarly, to strengthen the premise of the paper, the authors should determine an estimated number of CNEs that have diverged across the genome. This would require some assumptions about power and regions analyzed, but would be an important guideline for the number of regions specifically affecting limb development.

We agree with the importance of power estimations. However, they require an accurate knowledge of when the trait was lost in the evolution of the different lineages, as sequence divergence - and thus the strength of the signal - is proportional to the amount of time since the genomic regions have been evolving neutrally. Unfortunately, given the sparsity of the fossil record, it is not well known when limbs in snakes and functional eyes in subterranean mammals were lost. Nevertheless, we can provide a rough estimation based on available knowledge:

For snakes, our phylogenetic tree with branch lengths corresponding to the number of substitutions per neutral site shows that the boa and python terminal branches are ~0.07 subs/site long. Since the ancestor of all extant snakes was most likely already limbless, neutral evolution must have been ongoing for an even longer period. If we assume neutral evolution rate proportionally to 0.1 subs/site, we would have a power of ~75% to detect neutrally evolving regions based on sequence divergence at a precision of 90% (estimated based on simulations, shown in our previous study; Figure 3B in <https://academic.oup.com/mbe/article/33/8/2135/2579415>).

Compared to limb loss, eye degeneration in subterranean mammals evolved more recently. The earliest known fossils in the blind mole rat lineage are dated to ~20-24 Mya, which would correspond to 0.06-0.07 sub/site on the branch leading to this species. If we assume a similar rate of neutral divergence for the other subterranean mammals, where the timing of eye degeneration is not well known, we would have a power of 40-50% to detect neutrally evolving regions based on sequence divergence at a precision of 90% (estimated based on simulations, shown in our previous study; Figure 3B in <https://academic.oup.com/mbe/article/33/8/2135/2579415>).

Importantly, not all neutrally evolving regions will correspond to limb or eye regulatory elements. For example, limb loss is not the only phenotypic change in snakes and our new analysis (point 5 of reviewer 3) shows that snake-diverged CNEs also significantly overlap eye-regulatory data, likely because ancestral snakes also experienced a reduction in eye sight.

3) The limbless analysis uses a normalization of divergence and then a Z-score cutoff. Is the

seqID statistic normally distributed? That is an assumption of using the Z-score. It's not clear whether this strategy is statistically sound.

The reviewer is correct to assume that the %identity values are not normally distributed (they are also bounded between 0 and 1). However, the Z-score simply measures the number of standard deviations that an observation is above or below the mean. As written on Wikipedia (https://en.wikipedia.org/wiki/Standard_score) "They [Z-scores] are most frequently used to compare an observation to a standard normal deviate, though they can be defined without assumptions of normality."

The Z-test uses Z-scores to infer a *p*-value, which we did not do.

Instead, according to your valuable suggestion above, we have used *fdrtool* to estimate the false discovery rate for our Z-scores, as described now in the results and methods sections.

4) Rather than choosing an arbitrary cutoff of $Z < -3$, the authors should apply some type of false discovery rate correction or alternatively examine the tail of the distribution and measure enrichment along it. This opportunity is missed.

Following up on the first point, we now show that our chosen Z-score cutoff of -3 corresponds to a rather low false discovery rate of 1.186%.

5) The manuscript does not sufficiently explain how regions were scored as an ATAC-seq peak, and how tissue-specificity of a peak was inferred.

We apologize that our explanation of how we determined ATAC-seq peaks and identified tissue-specific regions of open chromatin was not clear enough. To correct this, we added more details to page 6 (Results section) and improved the description of the tools we used to call peaks (MACS2) and identify tissue-specificity (DiffBind) in the methods section:

Pg.6, results:

"We used MACS2³⁹ to identify genomic regions of open chromatin (peaks) and determined tissue-specific peaks with DiffBind⁴⁰."

Pg.18, methods:

"We used MACS2³⁹ to identify discrete peaks of enriched ATAC-seq signal in each sequencing library, using the mappable portion of the tegu genome (Hotspot *getMappableSpace.pl* script;¹¹¹) and using the shifted reads as input (MACS2 parameters '*--nomodel --shift -50 --extsize 100*'). Tissue-specificity of ATAC-seq peaks was determined using DiffBind⁴⁰, providing the peak coordinates for each of the biological replicates of all tissues profiled as input, plus the mapped and shifted sequencing reads (parameters '*method=DBA_EDGER, bFullLibrarySize=FALSE, bSubControl=FALSE, bTagwise=FALSE*'). All peaks identified with a log₂ fold change equal or greater than 1 in one tissue compared to all others were selected as tissue-specific."

6) Pg 6. Snake-diverged CNEs significantly overlap limb regulatory elements. 933 out of the total 5000 odd CNEs in total show overlap with any limb regulatory. Where do these 933 regions rank in terms of sequence divergence? It will be interesting to see if they are near the top or span the entire range, considering the ZRS enhancer ranks 4205 among the diverged CNEs.

We thank the reviewer for bringing up this very relevant point. The 933 snake-diverged CNEs that overlap limb regulatory data are broadly distributed among all 5,439 diverged CNEs, as we now show in Supplementary Figure 7. Furthermore, the distributions of FDR values of the 933 diverged CNEs that overlap limb elements and the remaining diverged

CNEs are quite similar, probably reflecting our relatively stringent cut-offs. We also analysed the CNEs diverged in subterranean mammals, comparing those that overlap eye regulatory data and the remaining diverged CNEs, and that also showed no major difference. This new data is shown in a new Supplementary Figure 7. Furthermore, we have added the Z-scores and FDR values for snake-diverged CNEs, and FDR values for the CNEs diverged in subterranean mammals to Supplementary tables 4 and 10, respectively.

We also added this information to the main text:

Pg.7, results:

“In total, 933 snake-diverged CNEs, widely distributed through the FDR range, overlap at least one of the tested limb regulatory datasets (Supplementary Figure 7, Supplementary Table 4).”

Pg.10, results:

“In total, we found 575 diverged CNEs, widely distributed through the FDR range, that overlap at least one of the tested eye regulatory datasets (Supplementary Figure 7, Supplementary Table 12).”

7) Pg 18. Detection of CNEs diverged in snakes: The boa and python snake lineage is considered to represent a single evolutionary loss event. Since their divergence from the MRCA, the authors argue the sequences are neutrally evolving. Given that, why do they not use them as two loss events? This would also mean they can use their Forward Genomics method to detect CNEs associated with the trait, meaning they have a consistent method across the two traits they study (limb and eye degeneration).

We cannot consider the two snake lineages as two independent loss events because their common ancestor was already limbless (Brandley et al. 2008; Caldwell 2003). This means that divergence in genomic regions that are associated with limb formation in limbed species also occurred before both snakes split. This is evident by the many deletions and substitutions that are shared between both snakes (e.g. Supplementary Figures 1 and 2) and many CNEs are entirely deleted in both snakes.

As we are interested in identifying regulatory elements that are associated to the loss of limbs, we conservatively computed the Z-score by using the sequence identity value of the least diverged snake. It would not be appropriate to apply our Forward Genomics approach to data that represents just a single loss lineage as this violates the key assumption of the method that the data comes from lineages where phenotype loss occurred independently.

8) Pg10. CNE divergence in TF binding sites: To understand if the diverged CNEs are associated with lost TFBS, the authors compare the motif score distribution for diverged CNEs against background CNEs. But the choice of CNEs are already informed by regulatory data from ATAC-seq and other experiments as they consider only CNEs that they detect as overlapping with regulatory data. This is a biased set, and perhaps a better test would be to perform the same analysis with all the CNEs they detect as being diverged.

We apologize that our rationale behind selecting only the diverged CNEs that overlap regulatory data was not clearly described. For CNEs, where it is unclear which functions they might have (in other words, snake-diverged CNEs that do not overlap limb regulatory data), it is not possible to select a specific set of TFs that may bind to these sequences. In contrast, for those CNEs that do overlap limb/eye regulatory data, it is reasonable to assume that known limb/eye-related TFs would bind. To make this clear in the text, we now write:

Pg.11, results:

“We first applied this strategy to analyze TF motif scores in the 933 snake-diverged CNEs that overlap limb regulatory data (Supplementary Table 4), reasoning that TFs relevant for limb development will bind to these sequences.”

“Next, we applied the same analysis to the 575 CNEs that are diverged in subterranean mammals and overlap eye regulatory data (Supplementary Table 12), reasoning that TFs relevant for eye development and function will bind to these sequences.”

The goal of this last results section is to test whether sequence divergence affects TF binding sites and, thus, the regulatory potential of the CNEs, or whether divergence occurred outside the binding sites, leaving regulatory function largely intact. To improve the clarity in the text, we added:

Pg.10, results:

“On the other hand, if regulatory function is largely preserved in species exhibiting sequence divergence, we expect that mutations occurred predominantly outside of TF binding sites, which would be conceptually similar to preserving a protein sequence in a coding region with numerous synonymous changes.”

Furthermore, based on comment 3 of reviewer 3, who asked if limb TF binding sites got preferentially or selectively lost in snakes, we also repeated the analysis with TF motifs that were artificially created. This new analysis shows that CNE divergence in snakes and in subterranean mammals also results in a loss of binding sites for these artificial TFs, which is shown in a new Supplementary Figure 14 and described in the text:

Pg.11, results:

“It should be noted that this analysis does not imply that binding sites of limb and eye TFs are preferentially lost in snakes and subterranean mammals, respectively. Indeed, a similar pattern of absence of conservation in TF motifs was observed when scoring the 933 snake-diverged CNEs with eye TF motifs, and the 575 CNEs diverged in subterranean mammals with limb TF motifs (Supplementary Figure 14A-D). Furthermore, repeating this analysis with randomized TF motifs also reveals a similar binding site divergence pattern (Supplementary Figure 14E and F). This suggests that there is no selective loss of binding sites for limb or eye TFs, but rather an overall sequence divergence that affects the entire CNE. Altogether, these analyses imply that CNE divergence results in a large-scale loss of TF binding sites, indicative of divergence of regulatory activity.

Minor Concern:

9) Page 7. It is awkward to have the citations in superscript in the middle of the sentence. “compiled a limb regulatory network based on 60-63 and...” Perhaps “...based on previous studies...” and then cite at the end of the sentence.

We changed this sentence, and few others with the same problem:

Pg.8, results:

“To this end, we compiled a limb regulatory network based on previous studies (60-63) and...”

Reviewer #2 (Remarks to the Author):

The great morphological diversity between species has sparked much of biologists' interest into their underlying genetic mechanisms. Both protein-coding sequences, and their regulatory elements which control genes' spatiotemporal expression can account for the morphological changes. However, it has been proposed that due to the pleiotropic effect of gene sequences, cis-regulatory elements might have a greater contribution and have become recent focus of research. This paper selected two generally very conserved traits: limb and eye, which have been lost independently several times throughout vertebrate evolution. They identified 5,439 conserved non-coding elements (CNEs) that are specifically diverged in the snake, with 933 overlapped with putative limb regulatory datasets. These snake diverged CNEs are also significantly associated with known limb-patterning genes, supporting their important role in limb degeneration of snakes. Similarly, they identified 9,364 CNEs diverged in the subterranean mammals (mole rat species etc.) but conserved in other species, with 575 overlapped with tested eye regulatory datasets. Both together provide evidence for the divergence of cis-regulatory elements participating in the lost of certain complex traits. I think the genomic datasets generated from this study is very valuable, including one additional lizard's genome, Tegu lizard limb ATAC-seq, mouse eye ATAC-seq. However, the new insights that are offered from these datasets are not substantial. As it is expected that eye or limb related cis-regulatory elements would become diverged in species that have lost these traits. However, given there are even much larger numbers of other diverged CNEs (>80% of snake diverged CNEs, and >90% of subterranean mammals CNEs) that are not related to these traits, one may directly ask the potential reasons for such a high false positive rate. I have other detailed comments below:

1. I recognize that it is very difficult to directly identify limb enhancers and their targeting genes. The authors basically ask in the paper: how many snake-specific diverged CNEs are correlated with limb enhancers or limb patterning genes? I am curious about the question being asked the other way around, how many known limb enhancers (e.g., Monti et al. 2017) are overlapped with these snake diverged CNEs? As besides limb, many other traits may have undergone degenerative evolution (e.g., vision) in snakes, which may directly account for the rest CNEs that do not overlap with limb enhancers/genes.

Thank you for raising this important question, and for pointing us to the valuable dataset from Monti et al. While we have described the overlap of snake-diverged CNEs with known limb enhancers (such as those near Hox, Gli3, Gas1 etc.) in a paragraph on page 7, we did neither list which and how many limb regulatory elements actually overlap the snake-diverged CNEs nor which experimentally validated limb enhancers we looked at. To correct this, we have now updated Supplementary Table 4 to include a detailed list of which limb regulatory elements a diverged CNE overlaps (likewise for the eye study; Supplementary Table 12). We also added a new column to specify if a snake-diverged CNE overlaps one of the limb enhancers from the Monti et al. dataset. Finally, we added a new Supplementary Table (9) that lists all experimentally validated limb enhancers that we extracted from the literature, which CNEs overlap these enhancers and the Z-scores of the respective CNEs. This table shows that 5 of 58 validated limb enhancers that align in the tegu lizard genome overlap with snake-diverged CNEs. Also, many other enhancers overlap CNEs with some degree of divergence (negative Z-scores) but are not classified as diverged by our stringent -3 cutoff.

We next analysed how many limb enhancers obtained by Monti et al. overlap snake-diverged CNEs. We found that 255 of 1656 (15.4%) enhancers that align to the tegu lizard genome overlap with snake-diverged CNEs. Furthermore, we included the limb enhancers predicted by Monti et al. 2017 in the enrichment tests, now shown Figure 2B. Indeed, the test shows that snake-diverged CNEs overlap this dataset with the highest significance of

all tests, probably reflecting the quality of the regulatory regions compiled by Monti et al. We have added this information to the manuscript:

Pg.6, results:

“Importantly, we observed the most significant overlap with a dataset of limb enhancers that was obtained by integrating many limb regulatory datasets and conserved transcription factor binding sites⁵⁶. “

As suggested by the reviewer, we further asked the question the other way around, and investigated if limb regulatory elements overlap snake-diverged CNEs significantly more often regulatory elements active in non-limb tissues. As shown in a new Supplementary Figure 6, this is indeed the case. We write next:

Pg.6, results:

“Finally, asking the question the other way around, we also found that limb regulatory elements overlap snake-diverged CNEs significantly more often than regulatory elements active in non-limb tissues (Supplementary Figure 6).”

We further used the high-quality limb enhancer dataset from Monti et al. to discuss about pleiotropy of gene regulatory elements:

Pg.13, discussion:

“For example, of the 5,786 CNEs that overlap limb enhancers obtained by integrating several regulatory datasets and conserved TF binding sites⁵⁶, only 315 (5.4%) have Z-scores lower than -3, corresponding to significant sequence divergence in snakes. While these 5,786 limb enhancer-overlapping CNEs overall have lower Z-scores compared to the remaining CNEs (Supplementary Figure 16, two-sided Wilcoxon rank sum test $p < 2.2e-16$), indicating some degree of sequence divergence, 36% of them show no evidence for divergence in snakes (Z-score ≥ 0).”

Finally, as suggested by the reviewer, other traits regressed in the evolution of snakes, which can explain divergence of other non-limb CNEs in snakes. To substantiate this point, we have strengthened the evidence that eye regulatory elements are also diverged in snakes by showing a significant enrichment of snake-diverged CNEs to many eye regulatory elements (in particular regulatory elements active in the retina). In total, there are 358 snake-diverged CNEs that overlap eye regulatory elements (only 92 of them also overlap limb regulatory data and could therefore be pleiotropic regulatory elements). This shows that the remaining snake-diverged CNEs that are not potential limb regulatory elements can overlap regulatory elements that are related to other traits in snakes. The results of this analysis are now included in Supplementary Figure 8, Supplementary Table 10, and in the text.

Pg.13, discussion:

“For example, we found that snake-diverged CNEs significantly overlap regulatory elements active during normal eye development (in particular retina development; Figure 2B, Supplementary Figure 8; Supplementary Tables 8 and 10). Together with the loss of opsins early in snake evolution^{96,97}, this enrichment is consistent with a possible subterranean origin of the crown snake lineage⁹⁸.”

2. In snakes, it is known from fossil data that forelimbs have become lost before the hindlimbs. Do the snake diverged CNEs or limb-ATAC peaks have a different distribution regarding their overlaps with forelimb and hindlimb genes/enhancers?

This is an interesting and relevant question. As forelimb loss in snakes is older than hindlimb loss, it could be expected that forelimb enhancers located near genes preferentially up-regulated in the forelimbs would be more diverged than those regulating hindlimb development. In addition, since hindlimbs and the genital system share regulatory elements (Infante et al. 2015), one could also expect less divergence in hindlimb enhancers.

To test this, we obtained genes up-regulated in mouse fore- or hindlimb buds as well as H3K27ac ChIP-seq-predicted enhancers preferentially active in fore- or hindlimbs (both datasets from Cotney et al. 2012) and computed the significance of the overlap between snake-diverged CNEs. We found that the 5,439 snake-diverged CNEs have no preferential enrichment with fore- or hindlimb upregulated genes or with fore/hindlimb enhancers. Similar results were obtained with the 933 diverged CNEs that overlap (any) limb regulatory elements. The absence of a detectable genomic signature for the evolutionary trajectory of the limb reduction process in the snake lineage likely reflects the accumulation of numerous neutral mutations over millions of years after limb reduction/loss happened in ancestral snakes.

These results are shown in a new Supplementary Figure 9A and were added to page 7, results:

“Consistent with widespread divergence of limb regulatory elements, we found that snake-diverged CNEs have no preferential association with genes upregulated in either fore- or hindlimbs, and no preferential overlap with fore- or hindlimb enhancers (Supplementary Figure 9A).”

Based on this comment, we went further and also tested whether the CNEs diverged in subterranean species are preferentially associated with lens- or retina-related genes and lens- or retina-specific ATAC-seq peaks. To this end, we obtained a list of 162 and 485 genes that give exclusively lens and retina phenotypes, respectively, when knocked-out in mice. We also used our lens- and retina-specific ATAC-seq peaks. We observed that CNEs diverged in subterranean mammals are preferentially associated with lens-related genes and with lens-specific ATAC-seq peaks compared to retina-related genes and retina-specific peaks. This genomic signature is consistent with observations that the lenses of subterranean mammals are highly degenerated, likely because light-focusing function of the lens is not necessary, while these species possess a retina (thinner with less photoreceptors but still exhibiting the typical layered structure), presumably to regulate the circadian rhythm. This shows that such differential divergence signatures can be found if the trait loss is more recent. We show this new analysis in a Supplementary Figure 9B, and also in discuss these findings in the text:

Pg.10, results:

“Interestingly, the CNEs diverged in subterranean mammals are preferentially associated with lens-related genes and with lens-specific ATAC-seq peaks, compared to retina-related genes and retina-specific peaks (Supplementary Figure 9B). This differential divergence signature is consistent with observations that the lenses of subterranean mammals are highly degenerated, likely because the light-focusing function of the lens became dispensable, while the reduced but normally-structured retina of these species is likely still involved in regulating the circadian rhythm^{92,93}.”

3. The putative mouse limb enhancers and their targeted genes have been recently characterized by Capture-C (Andrey et al. 2017). Among those ~400 limb-patterning genes and their inferred regulatory elements, how many are overlapped with the snake diverged CNEs?

We had already tested if snake-diverged CNEs were enriched for the overlap with the Andrey et al. 2017 capture C dataset of mouse limb genes and limb enhancers. We found that the limb genes are significantly enriched (as already shown in Figure 2A) and that the limb enhancers have a corrected p -value of 0.05035 (Supplementary Table 8, now included in Figure 2B), and are therefore slightly above the 0.05 threshold.

The absolute overlap of genes and regulatory elements characterized by CaptureC is as follows:

Of the 439 limb genes in the tegu lizard genome, 158 (36%) have at least one snake-diverged CNEs in their regulatory domain. This shows that the snake-diverged CNEs are not clustered around only a few limb genes, but are rather associated with many different genes. We also updated the text to include such information:

Pg.5, results:

“The analysis shows that snake-diverged CNEs are significantly enriched near genes that are involved in limb development and linked to congenital limb malformations³⁴ (158 of 439 genes, Figure 2A, top panel; Supplementary Table 5).”

Of the 496 CaptureC limb-specific enhancers that align to the tegu genome, 33 (6.6%) overlap a total of 45 snake-diverged CNEs. We have now added a detailed list of all the datasets that each snake-diverged CNE overlaps in Supplementary Table 4 and added the list of snake-diverged CNEs that are in the regulatory domains of the capture C genes (Supplementary Table 5).

4. Although the patterns of eye and limb loss regarding the cis-regulatory elements are similar, their experienced time-scale are different: the eye loss is much more recent than the limb loss of snakes. I wonder if it is possible to test using the current datasets that protein coding genes and cis-regulatory elements may have different contributions at different evolution stage. Or at least this should be discussed, otherwise, the two datasets and the conclusion look redundant in the paper.

We intended to combine the analysis of limb and eye loss in a single manuscript not to create redundancy, but rather to investigate if divergence of the *cis*-regulatory landscape is a general feature associated with the loss of complex phenotypes instead of being a feature specific to either limbs or eyes.

The question of the differential contribution of gene or regulatory divergence is an interesting question, which points out an important difference between limbs and eyes that we now discuss in a new paragraph. Interestingly, research from others and our group has shown that several eye-related genes were lost in subterranean mammals, despite eye loss being evolutionarily much more recent than limb loss. In contrast, *HoxD12* is the only limb-related gene that is reported to be lost in snakes. This shows that the different contribution of gene loss to phenotypic evolution is not caused by the age of phenotype loss, but rather by gene pleiotropy, as we describe in the text:

Pg.12, discussion:

“While divergence of cis-regulatory elements likely contributed to the loss of both limbs and functional eyes, the contribution of gene divergence noticeably differs between the two traits. In contrast to limbs, eyes consist of several unique tissues and cell types, such as lens or photoreceptor cells, which express a number of non-pleiotropic genes. In contrast, genes expressed in developing and adult limbs are often pleiotropic. This difference in pleiotropy predicts a differential contribution of gene loss to the regression of limbs and eyes. Indeed, despite the fact that eye degeneration happened evolutionarily much more recent compared to limb loss, several eye-specific genes diverged and

became inactivated in the subterranean mammals⁸³⁻⁸⁷, while in snakes only the loss of HoxD12 has been reported⁹⁴.”

5. Other minor points:

1) Page 3, I think here the previous studies have demonstrated that the retention of many limb-related enhancers is due to these enhancers' pleiotropy—that they are related to genital development as well. This should be mentioned in the introduction. As not only genes, but also regulatory elements can be also pleiotropic.

The reviewer is right that we only mention gene-pleiotropy in the introduction and discuss enhancer-pleiotropy only in the discussion. We have now added:

Pg.3/4, introduction:

“However, recent studies^{18,22,23} found that numerous other limb enhancers are nevertheless still conserved in snakes, despite limb reduction in this lineage dating back to more than 100 Mya²⁴, possibly due to pleiotropy of regulatory elements that drive expression in other non-limb tissues. Thus, it remains an open question whether phenotype loss is generally associated with divergence of the cis-regulatory landscape on a genome-wide scale”

2) Page 4, it is recommended to use BUSCO gene set to test the completeness of a draft genome. As the currently used ultraconserved elements were published long time ago, maybe either incomplete or not sufficient for the whole-genome coverage examination.

Thank you for this suggestion. We have run BUSCO using both the vertebrata and tetrapoda datasets. The tegu lizard genome gets the highest BUSCO scores of all species compared. The analysis is now included in Supplementary Table 2, and we modified the results and methods sections accordingly:

Pg.4, results:

“Our tegu assembly contains 197 of the 197 vertebrate non-exonic ultraconserved elements²⁷, and achieves a high BUSCO²⁸ score of 96.8%, showing an assembly completeness higher than that of all other sequenced reptiles (Supplementary Table 2).

Pg.16, methods:

“Second, we ran BUSCO²⁸ in genome mode using both the vertebrata_odb9 and tetrapoda_odb9 databases (creation date 2016-02-13 for both), which contain 2,586 and 3,950 highly conserved genes. The BUSCO score for the tegu lizard genome and for the genomes of other squamate reptiles are reported in Supplementary Table 2.”

3) Page 5, many other factors, like mutation coldspot or genetic drift could also lead to the formation of CNEs.

Thank you for this comment. We have revised the respective sentence to reflect that purifying selection may not be the underlying reason for CNEs:

Pg.5, results:

“...because evolutionary sequence conservation often implies purifying selection and thus function...”

Reviewer #3 (Remarks to the Author):

A major goal in evolutionary biology is identifying specific non-coding DNAs (including enhancers) responsible for the morphological change. In this paper, Hiller and colleagues use comparative genomics to determine a candidate set of enhancers that are associated with limb and vision loss in snakes and subterranean mammals respectively. First, the authors use multiple sequence alignment between many vertebrate genomes to identify conserved non-coding elements (CNE) that are specifically diverged in snakes and subterranean mammals. The authors then use various published gene expression and enhancer epigenomics data to show that indeed many of these diverged CNEs are enriched around limb/eye genes and overlap candidate limb/eye enhancers. Finally, they show that relevant TF binding sites are lost from diverged CNE, providing a potential mechanism for CNE deactivation during evolution.

The finding that limb and eye enhancers diverged/lost function in snakes and subterranean mammals has in principle been described by other groups before (e.g., (Infante et al. 2015; Partha et al. 2017)), so it is not unexpected per se. However, the value of the present work is that it provides an unbiased, purely sequence-based analysis framework for the identification of such divergent enhancers genome-wide, which should have wide applications for other phenotypes/species with a similar phenotype loss. The computational strategy itself is not particularly novel but represents a significant extension of a similar method by the same author (Hiller et al., NAR and Cell Rep, 2012). The new method focuses more on sequence divergence rather than a complete loss of non-coding DNA sequence, which is important given that enhancers often lose their function despite clearly recognizable sequence conservation (e.g., (Infante et al. 2015; Leal and Cohn 2016; Kvon et al. 2016)). Genome-wide list of candidate diverged CNEs will be a valuable resource for evolutionary biologists studying limb loss in snakes and vision loss in subterranean mammals. The authors also generated significant new experimental data (Tegu lizard whole genome sequencing and ATAC-seq data from lizard tissues) which, beyond their immediate application in the present manuscript, will be valuable resources for the scientific community. The paper is for the most part well written, although in some places it is fragmented and hard to follow - mainly because it is separated into independent limb and vision loss parts. This can be improved. Figures are very well designed and easy to understand.

While the presented analysis is rigorous and conclusions are sound, the paper could be significantly strengthened by addition of at least some experimental data empirically validating some of the findings. Specifically, it would be reassuring to see at least a few examples of candidate CNE that indeed lost/diverged their activity in an experimental setup to rule out the possibility that observed sequence changes are neutral with respect to CNE activity.

Major comments:

1) Empirical validation for diverged CNEs. It is nice to see that diverged CNE overlap putative limb/eye enhancers and are associated with limb/eye gene programs, especially because their identification was purely based on the analysis of sequence alignments and is independent of gene expression and epigenomics data. However, one would like to see functional validation of the CNE divergence in a model organism, for example using transgenic reporter assays. This would strongly support the main conclusion of the paper and rule out a possibility that these sequence changes are neutral with respect to CNE activity.

As the reviewer suggested, we performed experiments to test if the sequence divergence in CNEs in snakes can result in differences in regulatory activity. Since we do not have the ability to test enhancer activity of the sequences of different species in transgenic mice,

we tested CNE sequences of mouse, tegu lizard, and python in a dual luciferase reporter assay. To this end, we cloned the CNE sequences of each species into a pGL4.23[luc2/minP] firefly luciferase plasmid, and co-transfected with a pGL4.23[hRluc/SV40] renilla luciferase plasmid into NIH-3T3 cells.

As a positive control for the assay, we first synthesized, cloned, and measured the luciferase activity of the well-characterized *Shh* ZRS limb enhancer. By measuring the activity of the previously-tested mouse, green anole lizard, and python sequences, we found the expected loss of enhancer activity in the python, which recapitulates previous results (Supplementary Figure 15A).

Next, we selected 4 other snake-diverged CNEs which overlap our tegu limb ATAC-seq and published limb regulatory data. We synthesized and cloned the sequences of mouse, the tegu lizard, and the python, and tested them for regulatory activity. While for two of these four CNEs neither species' sequences showed an enhancer activity significantly different than the control vector (native pGL4.23 plasmid), the other two CNEs showed enhancing activity in these cells. For both, the python sequence drove significantly different expression levels compared to the mouse and tegu lizard sequences (Supplementary Figures 15B and C, Supplementary Table 22). For CNE011755 (located upstream of *Msx1*), the python sequence lost enhancing activity compared to the mouse and tegu lizard sequences, showing that divergence in the python sequence led to an impairment in enhancer function. Interestingly, for CNE107371 (located downstream of *Ebf2*), the python sequence drives significantly higher enhancer activity compared to the mouse and tegu lizard, showing, in this case, that sequence divergence in the python resulted in a release of the repressing activity that is present in the mouse and tegu sequences. Indeed, this region in the mouse is marked with chromatin repressor marks in fore and hindlimb buds (H3K27me3 ChIP-seq; Andrey et al. 2017).

These new experimental results are consistent with our computational results that sequence divergence in snakes resulted in a large-scale loss of TF binding sites, which can potentially affect regulatory activity, and are now described in Supplementary Figure 15, Supplementary Table 22, and in the manuscript:

Pg.12, results:

“To experimentally test if sequence divergence led to change in regulatory function, we compared the regulatory activity of CNE sequences of limbed and limbless species of four snake-diverged CNEs using luciferase enhancer assays. While two CNEs do not show enhancer activity in any species, the python sequence of the other two CNEs drives significantly different expression levels compared to the sequence of limbed species (Supplementary Figure 15). These experiments support the observation that phenotype loss is associated with the decay of the phenotype-specific cis-regulatory landscape.”

Pg.25, methods:

“Luciferase assays

We performed a dual luciferase assay to test diverged CNEs for regulatory activity. We synthesized the CNE sequences of mouse, anolis/tegu, and python of five different snake-diverged CNEs (including the CNE which overlaps the *Shh* ZRS limb enhancer), which were cloned into a pGL4.23[luc2/minP] vector (Promega) (Supplementary Table 22). The ZRS was used as a positive control for the assay, and the four other CNEs were selected based on their overlap with limb regulatory data and proximity to genes known to be involved in limb development.

Cell culture

NIH-3T3 cells (DSMZ) were cultured in DMEM medium (Thermo Fisher Scientific) supplemented with 10% fetal calf serum (Sigma), at 37°C, 5% CO₂ and 100% humidity. For detachment, cells were washed with PBS and treated with 0.05% Trypsin/EDTA (Thermo Fisher Scientific) for 3 minutes at 37°C. After incubation fresh medium was

added, cells were spun down at 140g for 5 minutes and re-suspended in fresh complete medium solution. 2500 NIH-3T3 cells/well were seeded into 384-well plates (Corning).

Transfection

Enhancer constructs were transfected 24 hours after seeding using FuGENE6 (Promega) as the transfection reagent, following manufacturer's instructions. In brief, FuGENE6 was mixed in OptiMEM (Thermo Fisher Scientific) and incubated for 5 minutes at room temperature. The plasmids with firefly (pGL4.23[luc2/minP]; Promega) and renilla (pGL4.73[hRluc/SV40]; Promega) luciferases were added to the mix in a ratio of 100:1 and, after an incubation period of 20 minutes at room temperature, the transfection complex was added to the cells. The optimal FuGENE/DNA ratio for NIH-3T3 cells was set to 8:1.

Luminescence readout

The read-out of the luminescence was done 24 hours after transfection, using the Dual-Glo Luciferase Assay System Kit (Promega) as substrate, according to manufacturer's instructions. For the read-out of the luminescence signal we used an Envision 2104 Multilabel reader (Perkin Elmer) with an ultra-sensitive luminescence 384-well aperture.

The assay for each CNE was repeated five times (five different plates), with a total of six technical replicates per plate for each CNE of each of the tested species. The ratio of firefly and renilla luciferases was normalized by the mean of the replicates of the empty control vector of the respective plate. Significance was assessed using a two-sided Wilcoxon rank sum test.“

2) Related to the previous comment. The authors show that their set of diverged CNEs includes ZRS limb enhancers which were shown to be degraded in the snake lineage. What about other known limb enhancers that diverged in snakes (e.g., HLEA and HLEB (Infante et al. 2015))? Was their algorithm able to find them?

We apologize for not making this information clearer before. The list of validated enhancers we looked at was mainly derived from Infante et al. 2015 (Supplementary Table S1 “Published limb enhancers”), with the addition of a few enhancers from VanderMeer et al. 2014 (ZPA and AER enhancers that drive expression of *Tcfap2b*, *Fgfr2*, *Sp8*, and *Arl13b*), the *Ptch1* LRM locus from Lopez-Rios et al. 2014, and 3 enhancers for *Hand2* genes from Monti et al. 2017.

From those 70 validated limb enhancers in mouse, 58 align to the tegu lizard genome, among them HLEB. We found that HLEB overlaps two CNEs that are most diverged in snakes; however, the divergence does not exceed our stringent cut-offs. We illustrate the divergence of these HLEB-overlapping CNEs in Supplementary Figure 17 and have now included this information in the main text.

Pg.13, discussion:

“While our genome-wide CNE screen detected sequence divergence signatures of the Island I enhancer, the sequence changes that underlie such partial enhancer activity differences may often be subtle without significantly increasing overall sequence divergence (for example, the pleiotropic HLEB limb and hemiphallus enhancer²² overlaps two CNEs with Z-scores of -1.3 and -2; Supplementary Figure 17; Supplementary Table 9).”

For HLEA, we find that it does not align to the tegu genome, consistent with Infante's observation that HLEA is not conserved in *Anolis*, gila monster, and boa genomes, and thus likely not conserved in reptiles.

The information of which validated enhancers overlap snake-diverged CNEs is now detailed in Supplementary Table 4. We also included an additional table (Supplementary table 9) listing all the experimentally validated limb enhancers that we investigated, those

that align to the tegu lizard genome, and the CNEs (and respective z-scores) that overlap each element.

3) Page 10, loss of TFBS. Is the specific loss of TF motifs in diverged CNE simply due to their overall poor conservation in snakes/subterranean mammals? An appropriate control would be comparing motif matches for limb TF motifs in eye enhancers and vice versa, for eye TF motifs in limb enhancers. If they also show decay, the motif loss is likely due to overall CNE sequence divergence.

We thank the reviewer for this important point. The reviewer is absolutely correct that there is no tendency of selective or preferential loss of TF binding sites in snakes or subterranean mammals. To show this and to make this clear in the manuscript, we performed the suggested, and an additional analysis. First, as suggested, we scored snake-diverged CNEs for eye TF motifs (and CNEs diverged in subterranean mammals for limb TF motifs) and show that putative eye/limb TF binding sites are also diverged in CNEs diverged in snakes and in subterranean mammals, respectively. Second, we generated artificial TF motifs by randomizing each motif, while preserving its information content. These randomized TF motifs result in a similar binding site divergence pattern. The rationale behind this second analysis is that (i) real TFs often have roles in more than one tissue and (ii) that different TFs can have similar motifs. Indeed, 14 TFs are in both our limb and eye TF set. Furthermore, of the remaining TFs, 28 TFs in our limb set have a similar binding motif to a TF in the eye set, and 30 TFs in our eye set have a similar binding motif to a TF in the limb set. Therefore, randomized motifs represent an unbiased set of motifs to test whether sequence divergence selectively affects real TF binding sites or not. Both tests now clearly show there is no selective binding site loss, but rather sequence divergence across the entire CNE.

We show these analyses in a new Supplementary Figure 14, and describe the analysis and the results in the text:

Pg.11, results:

“It should be noted that this analysis does not imply that binding sites of limb and eye TFs are preferentially lost in snakes and subterranean mammals, respectively. Indeed, a similar pattern of absence of conservation in TF motifs was observed when scoring the 933 snake-diverged CNEs with eye TF motifs, and the 575 CNEs diverged in subterranean mammals with limb TF motifs (Supplementary Figure 14A-D). Furthermore, repeating this analysis with randomized TF motifs also reveals a similar binding site divergence pattern (Supplementary Figure 14E and F). This suggests that there is no selective loss of binding sites for limb or eye TFs, but rather an overall sequence divergence that affects the entire CNE. Altogether, these analyses imply that CNE divergence results in a large-scale loss of TF binding sites, indicative of divergence of regulatory activity.”

Pg.24, methods:

“To obtain limb TF motifs that have a different binding motif compared to eye TFs (and vice-versa), we computed pairwise similarity scores with TomTom¹⁴³ (parameters ‘-thresh 1 -dist ed’) and removed motifs of limb TFs that have a pairwise similarity score ≤ 0.01 to another motif of an eye TF. To obtain randomized limb and eye TF motifs, we implemented a column-wise replacement procedure: starting from our motif library, we sorted all 7,567 motif columns according to their information content into 20 bins. Each limb or eye TF motif was then resampled by replacing every column by a randomly chosen column from the corresponding information content bin.”

4) The authors speculate in the discussion about limb reduction in snakes having occurred

as a stepwise process, with gradual reduction of limbs. The data set presented here may offer the intriguing possibility to directly see the evolutionary signature of this gradual process. The substitution rate in individual degrading CNEs in snakes should in principle be directly correlated to the time from loss of function of that CNE. With suitable corrections for the evolutionary constraint of a given sequence in limbed vertebrates, the present data set should in principle offer the opportunity to determine the age of the loss of function in individual CNEs. That said, due to the relatively short length and limited number of substitutions in each CNE, this estimate will likely not be very accurate at the level of individual CNEs. However, it should be possible to model the expected distribution of substitution rates across the entire CNE set assuming an instantaneous loss of constraint on all limb enhancers at the root of the snakes. If the observed distribution is shifted compared to this model, i.e. if there are more than expected lower-substitution CNEs, this could be a direct indication of a gradual process. Depending on signal strength, it may even be possible to observe bi-/multimodal distributions corresponding to groups of enhancers associated with specific morphological features. If the data set is underpowered for this type of analysis, another possibility would be to ask if CNEs associated with genes responsible for basic/early processes (e.g. limb bud induction/differentiation) show an overall lower substitution rate than those involved in later limb developmental processes (bone elongation etc.) – if it was a gradual process, the enhancers should lose function in “reverse order” of the developmental events.

Thank you for this insightful comment. Indeed, detecting a genome-wide divergence signal that could be a signature of a gradual process of limb reduction would be very interesting, especially considering the many intermediate limb forms in the fossil record. As the reviewer observed, if limb reduction in snake evolution was a gradual process, it would in theory be possible to detect a divergence signature in the CNEs, with higher-diverged CNEs being associated with genes/enhancers from later developmental processes. Therefore, following your suggestion, we have determined the relative number of substitutions (observed number of substitutions divided by the sum of number of substitutions and identical bases) for each CNE in the branch leading to the boa/python ancestor. This value should reflect how long an individual CNE is evolving neutrally in the snake lineage, and has been added to Supplementary Table 4.

Then, we ranked the snake-diverged CNEs by the relative number of substitutions. Please note that we are comparing substitutions of one and the same phylogenetic branch, therefore no correction for evolutionary rates between different branches is necessary. We tested the top and bottom 250 diverged CNEs with the highest and lowest relative substitution number for preferential association with genes (enhancers) that are up-regulated (active) in different limb developmental stages. To this end, we obtained stage-specific transcriptomics data from Taher et al. 2012 and chromatin capture data from Andrey et al. 2017. However, as shown in the figure below, we did not observe a preferential association between CNEs with a higher relative substitution number (that likely evolved neutrally for a longer time) and genes or enhancers up-regulated or active in later developmental time points. The same results were obtained when repeating these tests with the top and bottom 500 or 1000 CNEs. Please note that we excluded short CNEs or CNEs with large deletions, where the relative number of substitutions cannot be accurately estimated.

Furthermore, we tested these CNE subsets for preferential association with fore- or hindlimb genes or enhancers (see reviewer’s 2 comment 2 above), and also did not find a preferential association between CNEs with a higher relative substitution number and fore- or hindlimb genes/enhancers.

The fact that relatively few genes or enhancers are selectively active in fore/hindlimbs or at different timepoints during limb development makes it likely hard to detect preferential associations with specific sets of snake-diverged CNEs. However, we believe that the main reason for the absence of a detectable signal is the long evolutionary history of

limblessness in snakes (over 100 My), during which numerous neutral mutations accumulated that obscured mutational signatures could have been present during the early stages of limb reduction and loss.

5) The section on page 12, starting with “Interestingly, our analysis...” was slightly confusing as it attempts to link the two phenotypes in this study, which up to this point are presented and analyzed independently. The way this is currently presented, the fact that snakes also have poor vision and have been speculated to have subterranean ancestors appears like a post-hoc explanation of unexpected observations. Overall this entire paragraph seems speculative and may not be needed. Alternatively, if loss of vision and/or underground lifestyle in snake ancestors seems important, it should be introduced earlier and considered in the Results section. But I don’t think this would help with the narrative, I view this as a peripheral observation that distracts from an otherwise clean study design and analysis strategy focused on the core question of this work.

We believe that it is important to represent and analyse the diverged CNE data in an unbiased fashion, because not only limbs/eyes but also other phenotypes have changed in snakes/subterranean mammals. Therefore, we would like to keep this paragraph. We have now strengthened the evidence that snake-diverged CNEs are enriched in eye regulatory elements. Our new analysis considers all eye regulatory datasets and shows that snake-diverged CNEs are enriched in particular in retina regulatory elements, which is consistent with the loss of opsin genes. We have added a new Supplementary Figure 8 and Supplementary Table 10 supporting this observation, and revised the text to:

Pg.14, discussion:

“For example, we found that snake-diverged CNEs significantly overlap regulatory elements active during normal eye development (in particular retina development; Figure 2B, Supplementary Figure 8; Supplementary Tables 8 and 10). Together with the loss of opsins early in snake evolution^{96,97}, this enrichment is consistent with a possible subterranean origin of the crown snake lineage⁹⁸.”

These vision-related enrichments in snakes also provide a ‘molecular fingerprint’ that supports a fossorial lifestyle in crown snakes, as suggested based on reconstructed skull morphologies in a recent paper in Nature Communications (da Silva et al. 2018; <https://www.nature.com/articles/s41467-017-02788-3>). Thus, we hope that this paragraph also contributes to the debate of origin of the snake lineage (terrestrial, subterranean/fossorial, or marine).

Given that this paragraph comes right before the last discussion paragraph, we hope it is allowed to discuss important aspects other than our core message. To improve clarity in the transition from the previous to this paragraph, we rephrased the lead-in sentence to:

“Apart from divergence signatures related to limb loss and eye degeneration, our analysis also detected signatures related to other phenotypes that changed in snakes and vision-impaired subterranean mammals.”

Minor comments:

1) Abstract, last sentence: "Together, our results provide the first evidence that genome-wide decay of the phenotype-specific cis-regulatory landscape is a hallmark of lost morphological traits." - The work does not represent the "first evidence." See above papers (especially

Partha et al. 2017) showing the same trend for eye enhancers in mole rats and moles. Please tone down this sentence.

The reviewer is right that Partha et al. analysed a set of eye enhancers and showed sequence divergence in subterranean mammals; however, they did not perform an unbiased, genome-wide analysis; instead they limited the analysis to genomic loci around eye transcription factors. Nevertheless, we apologize if our statement appeared overly strong and therefore we have changed the sentence to:

“Together, our results provide evidence that genome-wide decay of the phenotype-specific cis-regulatory landscape is a hallmark of lost morphological traits.”

Furthermore, we have revised another two sentences:

Pg.2, introduction:

“Our analyses provide genome-wide evidence that divergence of the phenotype-specific cis-regulatory landscape is a hallmark of lost morphological traits.”

Pg.13, discussion:

“In summary, our study presents a comprehensive picture of non-coding changes that may have contributed to limb loss and eye degeneration...”

2) Please include line numbers in the next submission.

Line numbers have been added.

Reviewers' comments:

Reviewer #1 (Remarks to the Author):

A Major Concern remains with the statistical treatment of their analysis. The impact of this unaddressed, potential issue is that the false discovery rate is strongly underestimated.

The authors use a statistical procedure that makes an assumption about the null distribution of their seqID statistic. They do not know the distribution under the null and they don't attempt to characterize or simulate it. They should not rely on parametric assumptions. Also, fdrtool requires an expected distribution which in this case would be the normal distribution. In using Z-scores for a statistic that is not normally distributed under the null hypothesis has the potential to drastically skew false discovery rates. The authors must:

- 1) Determine the null distribution using an appropriate control set of species.
- 2) Use that new null distribution to provide better-informed false discovery rates.

Without this correction, the authors are reporting false discovery rates for the rejection of the null hypothesis that their statistic is normally distributed. However, the subject of the paper is rather test whether their regions are deviants from the null hypothesis of *non-convergence*.

- 3) The genomic and computational data generated in this study will be highly useful to the developmental community. In order for these groups to follow up with experiments to validate these regions the authors need to provide their data, including lists of all regions, all scores, and new sequences generated.

Reviewer #2 (Remarks to the Author):

During this round of revision, the authors performed additional analyses and addressed all my previous questions. They analyzed the limb enhancer dataset from Monti et al. 2017 and found that CNEs that specifically diverged in snakes have a significant overlap with the Monti et al. dataset, also than any other non-limb regulatory sequences. This result added new power to their previous ones. They also did not find differential enrichment of snake-diverged CNEs with fore- or hind- limb upregulated genes/enhancers, probably due to the long-term evolution that wipe out the signature of different orders of limb loss in snakes. They also discussed about the different contribution of cis-regulatory elements and gene sequences to the loss of trait. Overall, I think this work is ready to be accepted for publication.

Reviewer #3 (Remarks to the Author):

The revised version addresses all of my concerns.

I appreciate the thoughtful revisions, and in particular the effort the authors have put into performing additional experimental work and computational analyses, which have significantly strengthened the manuscript.

Point by point response

Reviewer #1 (Remarks to the Author):

A Major Concern remains with the statistical treatment of their analysis. The impact of this unaddressed, potential issue is that the false discovery rate is strongly underestimated.

The authors use a statistical procedure that makes an assumption about the null distribution of their seqID statistic. They do not know the distribution under the null and they don't attempt to characterize or simulate it. They should not rely on parametric assumptions. Also, fdrtool requires an expected distribution which in this case would be the normal distribution. In using Z-scores for a statistic that is not normally distributed under the null hypothesis has the potential to drastically skew false discovery rates. The authors must:

- 1) Determine the null distribution using an appropriate control set of species.
- 2) Use that new null distribution to provide better-informed false discovery rates.

Without this correction, the authors are reporting false discovery rates for the rejection of the null hypothesis that their statistic is normally distributed. However, the subject of the paper is rather test whether their regions are deviants from the null hypothesis of *non-convergence*.

As suggested, we have now determined the null distribution using simulations. Specifically, we simulated the evolution of an ancestral genome with annotated genes and 38,090 CNEs along the real phylogeny. We used the obtained simulated CNE data from all species to calculate Z-scores, as done for the real data. To estimate the null distribution of Z-scores, all CNEs evolved under selection in all species, including the snakes. Thus, any preferential sequence divergence in snakes (Z-scores < -3) is due to random chance alone. These two null distributions are now shown in a revised Figure 1D panel:

Since 40 out of 38,090 simulated CNEs have Z-scores < -3 , we estimate a false positive rate of 0.105% and therefore expect 172.7 false positives in the entire set of 164,422

real CNEs. Thus, 172.7 of the 5,439 snake-diverged real CNEs are expected to be false positives, which corresponds to a FDR of 3.17%.

As suspected by the reviewer, this new FDR estimate is indeed higher than the 1% estimated with `fdrtool`; however, both estimates show that our dataset overall has an acceptably low false-discovery rate.

The simulation and the new FDR estimate is described in the methods and presented in the Results.

We sincerely thank the reviewer for this helpful comment.

3) The genomic and computational data generated in this study will be highly useful to the developmental community. In order for these groups to follow up with experiments to validate these regions the authors need to provide their data, including lists of all regions, all scores, and new sequences generated.

We apologize if it was not clear from our previously-revised manuscript that the data is already available online or in the Supplementary Tables.

We have now also added the sequence divergence values (global and local percent identity values) of all CNEs in both sets (not only the diverged ones) to https://bds.mpi-cbg.de/hillerlab/CNE_Divergence/. The new files are named:

- limbStudy.CNE.global.percentID.txt
- limbStudy.CNE.local.percentID.txt
- eyeStudy.CNE.global.percentID.txt
- eyeStudy.CNE.local.percentID.txt

We now also provide the percent identity values of all simulated CNEs at https://bds.mpi-cbg.de/hillerlab/CNE_Divergence/ in the file `limbStudy.simulatedCNE.global.percentID.txt`.

As stated in the “Data availability” section of the manuscript, all other data produced in this study is already publicly available. Below we provide a detailed description of the datasets and their location.

All raw sequencing data is publicly available on NCBI SRA platform, under the project accession numbers PRJNA473319 (tegu lizard genome and transcriptome sequencing data), PRJNA481520 (tegu lizard ATAC-seq data), and PRJNA481646 (mouse ATAC-seq data).

The remaining datasets are provided on https://bds.mpi-cbg.de/hillerlab/CNE_Divergence/ and in the Supplementary Tables:

1. Files publicly available on https://bds.mpi-cbg.de/hillerlab/CNE_Divergence/ are:
 - Tegu lizard genome; file name `teguLizardGenome.fa`
 - Repeat masker annotation for the tegu lizard genome; file name `teguLizard.RM.out`
 - Multiple whole genome alignment (maf format) used in the limb study; file name `tegu29wayAlignment.maf.gz`
 - Tegu lizard gene annotation; file name `teguLizard.genes.gp`
 - Tegu lizard genes with human orthologs; file name `teguLizard.genes.WithHumanOrthologs.gp`
 - Full list and coordinates of conserved non-coding elements (CNEs) computed for the limb study; file name `teguCNEs.bed`

- Multiple whole genome alignment (maf format) used in the eye study; file name mouse24wayAlignment.maf.gz
- Full list and coordinates of conserved non-coding elements (CNEs) computed for the eye study; file name mouseCNEs.bed

At https://bds.mpi-cbg.de/hillerlab/CNE_Divergence/sequencing/ we provide all raw sequencing datasets as well as the MACS2 narrowPeak output of ATAC-seq peaks of all tissues and biological replicates sampled. This data includes scores of the ATAC-seq peaks.

2. Data available on Supplementary Tables:

- Coordinates (tegu lizard) of all 5,439 snake-diverged CNEs, their Z-score and FDR values, relative number of substitutions, and detailed description of the overlap with limb regulatory data (sup.Table 4)
- Coordinates (tegu lizard) of 439 limb related genes, and the snake-diverged CNEs associated with each gene (sup.Table 5)
- Coordinates (tegu lizard) of limb-specific ATAC-seq peaks (sup.Table7)
- Coordinates (mouse mm10) of 9,364 CNEs diverged in subterranean mammals, their FDR value, and detailed description of the overlap with eye regulatory data (sup.Table 12)
- Coordinates (mouse mm10) of the promoters of 64 eye-related genes lost in at least one of the vision-impaired subterranean mammals (sup.Table 13)
- Coordinates (mouse mm10) of eye E11.5-, lens E14.5- and retina E14.5-specific ATAC-seq peaks (supTable 16).

Reviewer #2 (Remarks to the Author):

During this round of revision, the authors performed additional analyses and addressed all my previous questions. They analyzed the limb enhancer dataset from Monti et al. 2017 and found that CNEs that specifically diverged in snakes have a significant overlap with the Monti et al. dataset, also than any other non-limb regulatory sequences. This result added new power to their previous ones. They also did not find differential enrichment of snake-diverged CNEs with fore- or hind- limb upregulated genes/enhancers, probably due to the long-term evolution that wipe out the signature of different orders of limb loss in snakes. They also discussed about the different contribution of cis-regulatory elements and gene sequences to the loss of trait. Overall, I think this work is ready to be accepted for publication.

We thank the reviewer for these kind words and the suggestions that helped to improve our manuscript.

Reviewer #3 (Remarks to the Author):

The revised version addresses all of my concerns.

I appreciate the thoughtful revisions, and in particular the effort the authors have put into performing additional experimental work and computational analyses, which have significantly strengthened the manuscript.

We thank the reviewer for these kind words and the suggestions that helped to improve our manuscript.

Reviewers' comments:

Reviewer #1 (Remarks to the Author):

The revision does not address my previous concern with potentially underestimated false discovery rates. My review specifically asked for control species, not simulations. The initial submission tested for departure from a normal distribution, which is clearly not the appropriate null model. The revision now tests departure from the null model of simulated sequences. I am afraid that sequences simulated with evolver are not guaranteed to return an appropriate null distribution of the percent identities. Sequence evolution is marked by extreme complexity that results in high variance and overdispersion of substitution rates. It is not clear that evolves models this. In short, the authors are now testing for departure from the null model of the evolver program, and they did not show, and I am not aware of, studies showing that evolved produces realistic overdispersion.

In the past review, I specifically asked the authors to examine the distribution from a control set of species that do not share a convergent trait. This is a reasonable null and would provide at least one reasonable null model against which to compute false discovery rates. I could also propose a permutation-based method on the real data that would return another more realistic null. Assuming that the authors have a matrix of percent identities, with CNEs as rows and species as columns, the authors could permute values within columns and compute a null distribution as before. (It might be best to first stratify CNEs into bins with similar overall evolutionary rates.)

A major stated goal of this study is to identify CNEs of importance to these developmental processes of limb and eye formation. If wet lab researchers are expected to follow these CNEs up with experiments, it is crucial to provide realistic false discovery rates as to help them plan the scale of experiments and which CNEs to prioritize. Without an appropriate null model, there remains the possibility that an FDR of 3% is a drastic underestimation.

Point by point response

Reviewer #1 (Remarks to the Author):

The revision does not address my previous concern with potentially underestimated false discovery rates. My review specifically asked for control species, not simulations. The initial submission tested for departure from a normal distribution, which is clearly not the appropriate null model. The revision now tests departure from the null model of simulated sequences. I am afraid that sequences simulated with *evolver* are not guaranteed to return an appropriate null distribution of the percent identities. Sequence evolution is marked by extreme complexity that results in high variance and overdispersion of substitution rates. It is not clear that *evolves* models this. In short, the authors are now testing for departure from the null model of the *evolver* program, and they did not show, and I am not aware of, studies showing that *evolved* produces realistic overdispersion.

In the past review, I specifically asked the authors to examine the distribution from a control set of species that do not share a convergent trait. This is a reasonable null and would provide at least one reasonable null model against which to compute false discovery rates.

We apologize for not mentioning in the previous revision that we had already computed a set of 616 real CNEs that are preferentially diverged in control species (the anole and dragon lizards), which we previously used as a control in the enrichment analyses. Following the reviewer's request, we now used this set to estimate the FDR. We believe that these two lizards are the most appropriate species pair to serve as controls because of their phylogenetic position: both lizards are sister species (like the two snakes), and both represent the direct sister group to the snakes (Figure 1B).

Making the assumption that all 616 CNEs diverged in anole and dragon lizards with Z-score cut-offs of -3 are evolving under purifying selection (thus, are false positives), we obtain an FDR of 11.33% (616 false positives among the 5,439 snake-diverged CNEs). However, this assumption is not realistic as it is not possible to rule out that these two lizards do not share a convergent trait and that some of these CNEs are related to phenotypic changes. Thus, this estimate corresponds to an upper bound of the FDR.

Nevertheless, an upper FDR bound of 11.33% would still be acceptable for experimental follow-up tests. This upper bound is also acceptable for genome-wide screens and several previous genomic studies have been using FDR cutoffs of up to 15% (e.g. <https://elifesciences.org/articles/25884> used a 15% FDR cutoff).

I could also propose a permutation-based method on the real data that would return another more realistic null. Assuming that the authors have a matrix of percent identities, with CNEs as rows and species as columns, the authors could permute values within columns and compute a null distribution as before. (It might be best to first stratify CNEs into bins with similar overall evolutionary rates.)

We performed a permutation-based method, exactly as suggested by the reviewer. We permuted the real sequence identity values between species (which are indeed columns in our matrix) and applied the same Z-score calculation. To obtain robust estimates, we repeated this test 10 times.

These permutation-based tests show that, on average, 109.4 CNEs have Z-scores < -3 in the snakes. This corresponds to an FDR of 2.01% (109.4 / 5439). The number of diverged CNEs detected in these 10 repetitions ranges from 90 to 124, corresponding to

FDR values of 1.654 to 2.2798%. Because these values fall between those obtained with `fdrtool` (1.186%) and our previous simulation (3.17%), we chose to be conservative and keep the higher 3.17% FDR estimate in the text.

We added the upper FDR bound of 11.33%, derived from the 616 CNEs diverged in the control species, to the text:

Results page 5:

“To estimate an upper bound of the FDR, we detected 616 CNEs that are preferentially diverged in the sister lineage of snakes comprising the anole and dragon lizards. Making the unrealistic assumption that all 616 CNEs evolve under purifying selection in both species and thus are false positives, an upper FDR bound for the snake-diverged CNEs is 11.33% (616 of 5,439 CNEs). This conservatively estimates that at most 11.33% of the snake-diverged CNEs may still evolve under purifying selection in snakes.

“

Methods page 22

“

We further used the set of 616 CNEs diverged in the green anole lizard and the dragon lizard to compute an upper bound of the false discovery rate. In this test, we assumed that all 616 CNEs actually evolve under purifying selection in both lizards, which is likely not the case as these CNEs could be related to phenotypic changes that are shared between these two species. Nevertheless, making this assumption, one would expect that 616 of the 5,439 snake-diverged CNEs are false positives, which corresponds to an upper bound of the FDR of 11.33%.

“

A major stated goal of this study is to identify CNEs of importance to these developmental processes of limb and eye formation. If wet lab researchers are expected to follow these CNEs up with experiments, it is crucial to provide realistic false discovery rates as to help them plan the scale of experiments and which CNEs to prioritize. Without an appropriate null model, there remains the possibility that an FDR of 3% is a drastic underestimation.

REVIEWERS' COMMENTS:

Reviewer #1 (Remarks to the Author):

The authors have made all of the requested discovery rate analyses, and the results support the strength of their inferences. All of my concerns have been satisfied, and I fully recommend acceptance of the manuscript for publication.